# The P-type ATPase transporter ATP7A promotes angiogenesis by limiting autophagic degradation of VEGFR2

Dipankar Ash [1,10], Varadarajan Sudhahar[1,2,10], Seock-Won Youn[1,3], Mustafa Nazir Okur[4], Archita Das[1], John P. O'Bryan[5,6], Maggie McMenamin[1,2], Yali Hou[1,2], Jack H. Kaplan[7], Tohru Fukai [1,8,2,11 ✉] & Masuko Ushio-Fukai [1,9,11 ✉]

VEGFR2 (KDR/Flk1) signaling in endothelial cells (ECs) plays a central role in angiogenesis. The P-type ATPase transporter ATP7A regulates copper homeostasis, and its role in VEGFR2 signaling and angiogenesis is entirely unknown. Here, we describe the unexpected crosstalk between the Copper transporter ATP7A, autophagy, and VEGFR2 degradation. The functional significance of this Copper transporter was demonstrated by the finding that inducible EC-specific ATP7A deficient mice or ATP7A-dysfunctional ATP7Amut mice showed impaired post-ischemic neovascularization. In ECs, loss of ATP7A inhibited VEGF-induced VEGFR2 signaling and angiogenic responses, in part by promoting ligand-induced VEGFR2 protein degradation. Mechanistically, VEGF stimulated ATP7A translocation from the trans-Golgi network to the plasma membrane where it bound to VEGFR2, which prevented autophagy-mediated lysosomal VEGFR2 degradation by inhibiting autophagic cargo/adapter p62/SQSTM1 binding to ubiquitinated VEGFR2. Enhanced autophagy flux due to ATP7A dysfunction in vivo was confirmed by autophagy reporter CAG-ATP7Amut -RFP-EGFP-LC3 transgenic mice. In summary, our study uncovers a novel function of ATP7A to limit autophagy-mediated degradation of VEGFR2, thereby promoting VEGFR2 signaling and angiogenesis, which restores perfusion recovery and neovascularization. Thus, endothelial ATP7A is identified as a potential therapeutic target for treatment of ischemic cardiovascular diseases.

---

[1] Vascular Biology Center, Medical College of Georgia at Augusta University, Augusta, GA, USA. [2] Charlie Norwood Veterans Affairs Medical Center, Augusta, GA, USA. [3] Department of Physiology and Biophysics, University of Illinois College of Medicine, Chicago, IL, USA. [4] Laboratory of Molecular Gerontology, National Institute on Aging, National Institutes of Health, Baltimore, MD, USA. [5] Department of Cell and Molecular Pharmacology and Experimental Therapeutics, Hollings Cancer Center, Medical University of South Carolina, Charleston, SC, USA. [6] Ralph H. Johnson VA Medical Center, Charleston, SC, USA. [7] Department of Biochemistry and Molecular Genetics, University of Illinois College of Medicine, Chicago, IL, USA. [8] Departments of Pharmacology and Toxicology, Medical College of Georgia at Augusta University, Augusta, GA, USA. [9] Department of Medicine (Cardiology), Medical College of Georgia at Augusta University, Augusta, GA, USA. [10] These authors contributed equally: Dipankar Ash, Varadarajan Sudhahar. [11] These authors jointly supervised this work: Tohru Fukai, Masuko Ushio-Fukai. ✉email: tfukai@augusta.edu; mfukai@augusta.edu

Angiogenesis, the process of new vessel formation from pre-existing vessels, plays important role in wound repair and restoring perfusion recovery and neovascularization in ischemic heart disease and peripheral vascular disease[1]. Vascular endothelial growth factor (VEGF), a key angiogenic growth factor, stimulates migration, proliferation, and capillary tube formation of endothelial cells (ECs) primarily through VEGF receptor type2 (VEGFR2), which promotes neovascularization and vascular regeneration[2]. VEGFR2 is a classic tyrosine kinase receptor, having an intracellular kinase domain that is activated upon ligand binding and dimerization, resulting in autophosphorylation, and activation of intracellular signaling, via MAPK and PI3K/Akt. Like other receptor tyrosine kinases, VEGFR2 was thought to signal from the cell plasma membrane after ligand-induced dimerization and activation[3]. However, a number of recent studies have challenged this notion by showing the importance of endocytosis and trafficking of VEGFR2 in regulating its signaling[2]. Upon VEGF stimulation, activated/dimerized VEGFR2 at the cell surface is internalized to early endosomes for activating signaling, but some fraction of the receptor is ubiquitinated and sorted to the lysosome for degradation, the remaining is recycled to the plasma membrane[4]. However, the molecular mechanisms that regulate VEGFR2 signaling via endocytosis and trafficking remain elusive.

The P-type ATPase transporter ATP7A is a key regulator of secretory Cu enzymes and of intracellular Cu levels[5,6]. Under basal conditions, ATP7A localizes at the trans-Golgi network (TGN) where it transports Cu to the secretary Cu enzymes, such as extracellular superoxide dismutase (SOD3) or the proenzyme of lysyl oxidase (Pro-LOX) required for LOX activation[5,6], which promotes tumorigenesis and metastasis[7]. It is also partially involved in VEGF or ischemia-induced angiogenesis in ECs[8,9]. In pathological conditions in which cellular Cu is elevated, ATP7A translocates from TGN to the plasma membrane to export the excess Cu. It has been shown that relocalization of ATP7A from the TGN is triggered not only by increased cytoplasmic Cu but also by non-metal stimulants such as insulin, NMDA, PDGF, and hypoxia[10]. The biological significance of ATP7A in vivo is underscored by Menkes disease which is caused by a loss-of-function X-linked mutation of ATP7A[11]. Of note, global ATP7A knockout mice are embryonic lethal due to vascular defects[12]. Mice carrying the X-linked blotchy ATP7A mutation (ATP7A[mut] mice) have a splice site mutation introducing a new stop codon at amino acid residue 794 with reduced Cu transport function and ATP7A expression. These mutant mice survive more than 6 months of age[5,13–15] and are well characterized to enable the study of the in vivo function of ATP7A in adult animals. Using ATP7A[mut] mice, we reported that vascular ATP7A protects against endothelial dysfunction in hypertensive[14] and in diabetic mice[16,17]. We also showed that ATP7A protein expression was markedly downregulated in diabetic vessels, which led to impaired endothelium-dependent vasorelaxation[16,17]. However, the role of endothelial ATP7A in VEGF-induced angiogenesis in ECs and postnatal angiogenesis in vivo has not been explored.

Autophagy is an evolutionarily conserved lysosomal degradation pathway mediating the clearance of damaged organelle and dysfunctional proteins within autophagosomes which fuse with acidic lysosomes[18]. The cargo has several components, including a selective autophagy adapter/cargo p62/Sequestosome 1 (SQSTM1) to detect and aggregate polyubiquitin proteins and bind to the integral autophagosome component LC3 for autophagosome generation[19]. It is degraded by lysosomal proteases to provide essential elements for maintaining cell metabolism[20]. However, the role of autophagy in angiogenesis is complex. For example, ATG5-mediated autophagy induced by nutrient deprivation is required for angiogenesis[21], while autophagy induced by glycolytic intermediate impaired angiogenesis[22]. Furthermore, there is a connection between autophagy, lysosomes, and the Cu transport proteins involved in regulating Cu metabolism. It is known that late endosomes/lysosomes import Cu via the ATP7A homolog, ATP7B, which moves to the lysosome from the TGN in response to Cu in hepatic cells[23]; induction of autophagy protects ATP7B deficient hepatocytes from excess Cu-mediated toxicity[24]; Cu accumulation in senescent mouse fibroblasts is due to the impaired interplay between ATP7A and the autophagic-lysosomal pathway[25]. However, the mechanistic linkages between ATP7A and autophagy in VEGFR2 signaling in ECs and neovascularization in vivo have not been characterized.

In this study, using inducible endothelial-restricted ATP7A knockout (iEC-ATP7A KO) mice obtained by crossing ATP7A[fl/fl] with tamoxifen-inducible iCdh5-CreERT2 mice[26] or using ATP7A dysfunctional ATP7A[mut] mice, we provide evidence that these ATP7A dysfunctional mice exhibit reduced ischemia-induced angiogenesis. We found an endothelial ATP7A-autophagy linkage, which is important in the regulation of VEGFR2 signaling. Loss of ATP7A in ECs promoted autophagosome formation and autophagic cargo/adapter p62/SQSTM1 binding to ubiquitinated VEGFR2, thereby accelerating VEGFR2 targeting to autophagolysosomes for degradation (in a Cu-independent manner), which in turn inhibits VEGF-induced angiogenesis. Thus, endothelial ATP7A functions to promote VEGFR2 signaling and angiogenesis by limiting autophagic degradation of VEGFR2, in addition to its canonical role in delivering Cu to Cu-dependent secretory enzymes, such as LOX. This role for ATP7A is required for restoring perfusion recovery and neovascularization in ischemic vascular disease.

## Results

**Endothelial ATP7A is required for post-ischemic revascularization.** To address the role of ATP7A in angiogenesis in vivo, we used the mouse hindlimb ischemia (HLI) model which induces ischemia by femoral artery ligation and excision, as previously reported[8,27]. Immunofluorescence analysis revealed that ATP7A protein expression was robustly increased and colocalized with CD31-positive ECs at day 14 in ischemic muscles compared to non-ischemic muscles (Fig. 1a). These results suggest that ATP7A is increased in angiogenic ECs in the HLI model. To determine the role of endogenous ATP7A for ischemia-induced angiogenesis, we used ATP7A[mut] mice with reduced ATP7A expression and Cu transport capacity[14]. Figure 1b, c showed that perfusion recovery and angiogenesis (CD31 positive capillaries), respectively, in response to hindlimb ischemia were significantly impaired in ATP7A[mut] mice compared to control, wild type (WT) mice. To determine the relative role of ATP7A in tissue-resident cells and bone marrow (BM) cells in post-ischemic neovascularization, we performed a BM transplantation experiment. Lethally irradiated WT mice transplanted with WT-BM or ATP7A[mut]-BM showed no significant difference in perfusion recovery after ischemic injury (Fig. 1d), suggesting that ATP7A in tissue-resident cells, but not in BM cells, is required for ischemia-induced neovascularization.

To determine the role of endothelial ATP7A in ischemia-induced angiogenesis on adult-onset in vivo, we generated inducible EC-specific hemizygous male ATP7A knockout (iEC-ATP7A KO) mice by crossing homozygous floxed females (ATP7A[fl/fl]) with mice expressing tamoxifen-inducible Cre recombinase under the control of the VE-cadherin promoter (ATP7A[+/Y]; Cdh5-CreERT2[+/−])[26] (Fig. 1e, f). Selective deletion of ATP7A in ECs was validated by qPCR analysis in ECs and vascular smooth muscle cells (VSMC) isolated from iEC-ATP7A KO mice (Fig. 1g, h). Although global ATP7A KO mice were embryonic lethal[12], iEC-ATP7A KO mice survived and

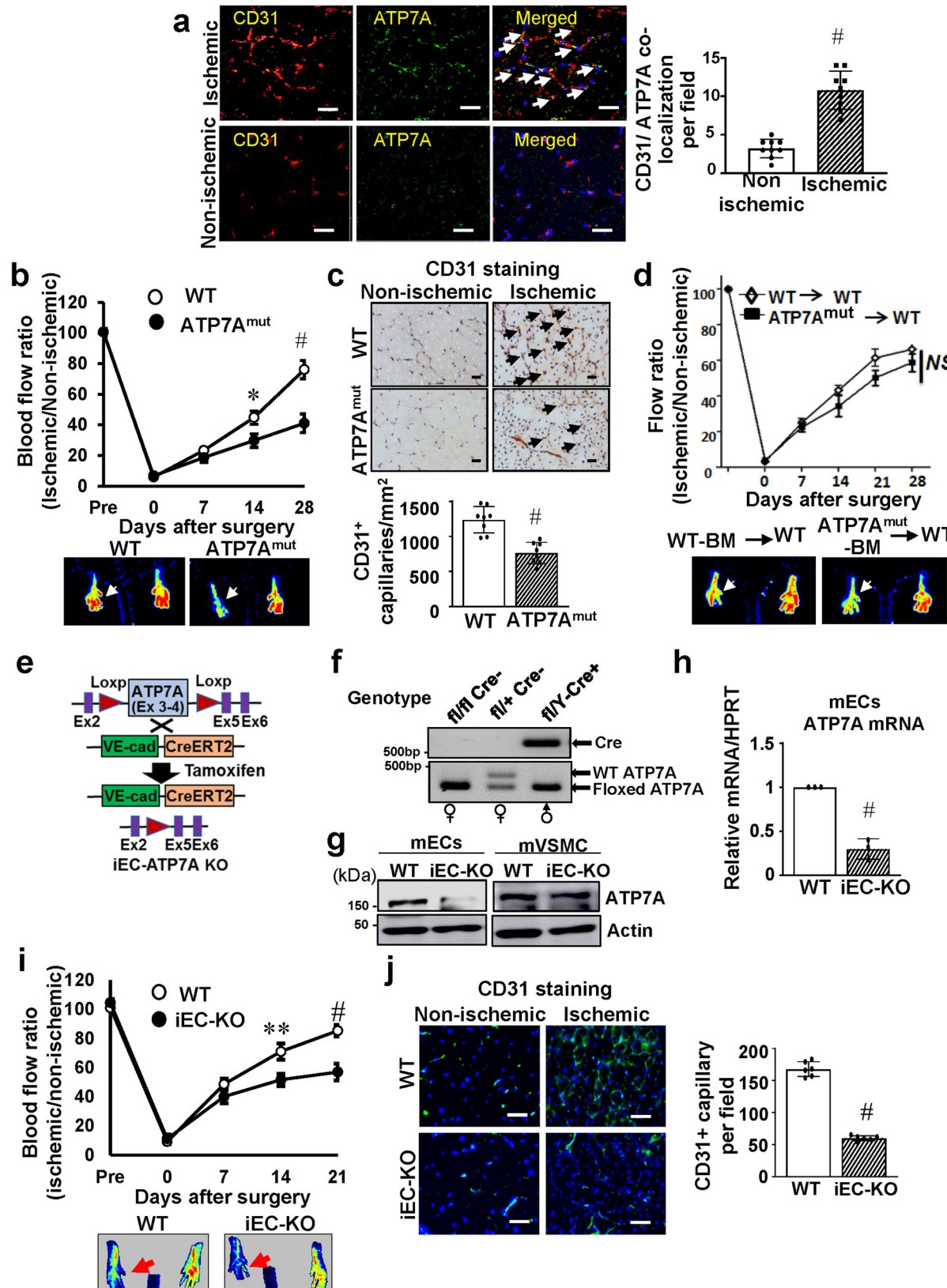

were indistinguishable from WT mice, displaying no obvious abnormalities. Blood flow recovery and CD31$^+$ capillary density in ischemic muscle were significantly impaired in iEC-ATP7A KO mice compared to WT mice (Fig. 1i, j). Thus, endothelial ATP7A plays an important role in post-ischemic neovascularization.

**ATP7A knockdown in ECs inhibits VEGFR2 signaling via promotion of VEGFR2 degradation in a Cu-independent manner**. We next examined the role of ATP7A in VEGF-induced angiogenic responses in primary cultured ECs. Figure 2a, b, using the modified Boyden chamber assay, showed that VEGF-induced EC migration was significantly inhibited in ECs isolated from

**Fig. 1 Endothelial ATP7A is required for ischemia-induced neovascularization. a** CD31 (red, Endothelial Cell (EC) marker) or ATP7A (green) staining or their colocalization (merged, white arrows) in ischemic and non-ischemic gastrocnemius muscles at day 14 post-surgery. Scale bars = 50 μm. The bar graph represents CD31-ATP7A co-localized cell numbers per field. $n = 6$, $\#p < 0.0001$ (two-tailed unpaired $t$-test). **b** Blood flow recovery as determined by the ratio of foot perfusion between ischemic (left) and ischemic (right) legs after hindlimb ischemia in Wild Type (WT) and ATP7A$^{mut}$ mice. $n = 5$, $*p < 0.05$, $\#p < 0.0001$ (two-way ANOVA followed by Bonferroni's multiple comparison analysis). Bottom panels show representative laser Doppler images (red arrows show ischemic foot) of legs at day 28 after ischemia in WT and ATP7A$^{mut}$ mice. **c** Capillary density (CD31 staining) in ischemic and non-ischemic muscles of WT and ATP7A$^{mut}$ mice at day 14. Scale bars = 50 μm. Bottom panels show their quantitative analysis (number of capillaries per mm square, $n = 8$, $\#p < 0.0001$ (two-tailed unpaired $t$-test). **d** Bone marrow (BM) from WT or ATP7A$^{mut}$ mice were transplanted to irradiated WT mice. After 6 weeks, mice were subjected to hindlimb ischemia and limb blood flow was measured. $n = 6$, NS = non-significant (two-way ANOVA followed by Bonferroni's multiple comparison analysis). The bottom panels show representative laser Doppler images on day 28. **e** Schematic representation generating inducible EC-specific hemizygous male ATP7A knockout (iEC-ATP7AKO) mice by crossing homozygous floxed females (ATP7A$^{fl/fl}$) with mice expressing the Cre recombinase located downstream of tamoxifen-inducible VE-Cadherin promotor (ATP7A$^{+/Y}$; Cdh5-CreERT2$^{+/−}$). **f** Representative gel images of genotyping from ATP7A$^{fl/fl}$Cre$^{−}$ (female), ATP7A$^{fl/+}$Cre$^{−}$ (female) and ATP7A$^{fl/Y}$ Cre$^{+}$ (male) mice. A similar pair of gels were resolved to genotype an average of 50 mice/breeding pair with $n = 3$ independent breeding pairs. **g, h** ATP7A protein (**g**) and mRNA (**h**) expression in ECs and aortic vascular smooth muscle cells isolated from WT and iEC-KO mice. $n = 3$ animals/group, $\#p = 0.0005$ (two-tailed unpaired t-test). **i** Blood flow recovery after hindlimb ischemia in WT and iEC-KO mice. The bottom panels show representative laser Doppler images on day 21. $n = 5$ animals/group, $**p = 0.01$, $\#p < 0.0001$ (two-way ANOVA followed by Bonferroni's multiple comparison analysis). **j** Capillary density (CD31 staining) in ischemic and non-ischemic muscles of WT and iEC-KO mice at day 14. Right panels show quantification for the number of capillaries per field in ischemic muscles. $n = 6$, $\#p < 0.0001$ (two-tailed unpaired $t$-test). Scale bars = 50 μm. Data are mean ± SEM.

ATP7A$^{mut}$ mice or HUVECs transfected with ATP7A siRNA. Of note, ATP7A siRNA had no effect on EC migration induced by sphingosine 1-phosphate (S1P), another potent angiogenic G-protein coupled receptor agonist (Supplementary Fig. 1), supporting the specificity of ATP7A siRNA action. Furthermore, the spheroid EC sprouting assay[28] showed that ATP7A depletion in ECs significantly inhibited the VEGF-induced increase in the number and length of sprouting from the spheroid (Fig. 2c). These results suggest that ATP7A is specifically involved in angiogenic responses induced by VEGF, but not by S1P in ECs.

To address the mechanism by which ATP7A regulates VEGF-induced angiogenesis, we examined the role of ATP7A in VEGFR2 signaling in ECs. ATP7A knockdown (KD) with siRNA significantly inhibited VEGF-induced VEGFR2 tyrosine phosphorylation and its downstream signalings such as phosphorylation of Akt and p38MAPK, by reducing total VEGFR2 protein (Fig. 2d). Importantly, ATP7A siRNA had no effects on S1P-induced phosphorylation of Akt or p38MAPK in ECs (Fig. 2e), supporting a specific role of ATP7A in VEGFR2 signaling. Since ATP7A depletion had no effect on VEGFR2 mRNA (Fig. 2f), we next examined whether the decrease in VEGFR2 protein in ATP7A-depleted ECs was due to enhanced degradation or reduced synthesis of VEGFR2. Figure 2g showed that VEGFR2 protein expression in the presence of cycloheximide, a eukaryotic protein translation inhibitor, was decreased after VEGF stimulation in a time-dependent manner in both control siRNA and ATP7A siRNA-transfected ECs. These results suggest that ATP7A knockdown in ECs promotes ligand-dependent VEGFR2 protein degradation, which was further confirmed using bovine aortic ECs transfected with two different ATP7A siRNAs (Supplementary Fig. 2). Of note, ATP7A KD in HUVECs had no effect on protein expression of neuropilin-1 (Nrp1), which functions as a VEGFR2 co-receptor[29] in HUVECs stimulated with VEGF (Supplementary Fig. 3a), nor on ligand-induced degradation of other receptor tyrosine kinases such as Insulin receptor 1β (IR1β) (Supplementary Fig. 3b), fibroblast growth factor receptor 1 (FGFR1) (Supplementary Fig. 3c) or VEGFR3 (Supplementary Fig. 3d) in HUVECs. Thus, ATP7A specifically regulates VEGFR2-mediated signaling and angiogenic responses in ECs.

We then examined if ATP7A KD-induced VEGFR2 degradation was a consequence of Cu accumulation induced by impaired Cu exporter (ATP7A) function using tetrathiomolybdate (TTM). TTM is an intracellular Cu chelator that has been used for the

treatment of Wilson Disease and in a number of clinical trials as an anti-tumor agent[30–35]. We found that TTM had no significant effect on either ATP7A KD-induced enhancement of VEGFR2 degradation (Fig. 2h) or inhibition on VEGFR2 signaling (Supplementary Fig. 4a) in HUVECs. The chelator efficacy of TTM was confirmed by showing that TTM almost completely inhibited VEGF-induced Cu-dependent lysyl oxidase (LOX) activity (Supplementary Fig. 4b), as well as CuCl$_2$-induced phosphorylation of ERK, which has been shown to be mediated through direct activation of MEK by Cu[36,37] (Supplementary Fig. 4c). These results suggest that loss of ATP7A in ECs inhibits VEGFR2 signaling via promoting VEGFR2 degradation in a Cu-independent manner.

**VEGF stimulation promotes ATP7A binding with VEGFR2.** To address the mechanism by which ATP7A stabilizes VEGFR2 protein, we next examined possible interactions between ATP7A and VEGFR2. A co-immunoprecipitation assay showed that VEGF stimulation in ECs rapidly increased ATP7A binding to VEGFR2 within 5 min in a time-dependent manner (Fig. 3a). In addition, to monitor in situ formation of ATP7A-VEGFR2 complex, we next used the Duolink proximity ligation assay (PLA)[38], where red puncta indicate positive staining that the epitopes of the target proteins are in close proximity (<40 nm). We found that both primary mouse ATP7A antibody and rabbit VEGFR2 antibody combined with secondary PLA probes dramatically increased the PLA positive red dots after VEGF stimulation in HUVECs (Fig. 3b). By contrast, single primary antibody only or IgG negative control did not display red dots (Fig. 3b). To confirm further their direct interaction, we performed co-transfection of myc-ATP7A and flag-VEGFR2 plasmids in Cos7 cells and found that myc-ATP7A, but not IgG, was co-immunoprecipitated with flag-VEGFR2 after VEGF stimulation (Fig. 3c). These results suggest the ligand-induced direct interaction between VEGFR2 and ATP7A in situ. In addition, co-localization analysis using confocal microscopy showed that VEGF stimulation promoted ATP7A translocation from TGN to the plasma membrane where it colocalized with VEGFR2 after 5 min, which was followed by co-internalization of ATP7A and VEGFR2 to the perinuclear region after VEGF stimulation for 15 min (Fig. 3d). Taken together, these data show that VEGF stimulation promotes direct binding of ATP7A with VEGFR2 in ECs.

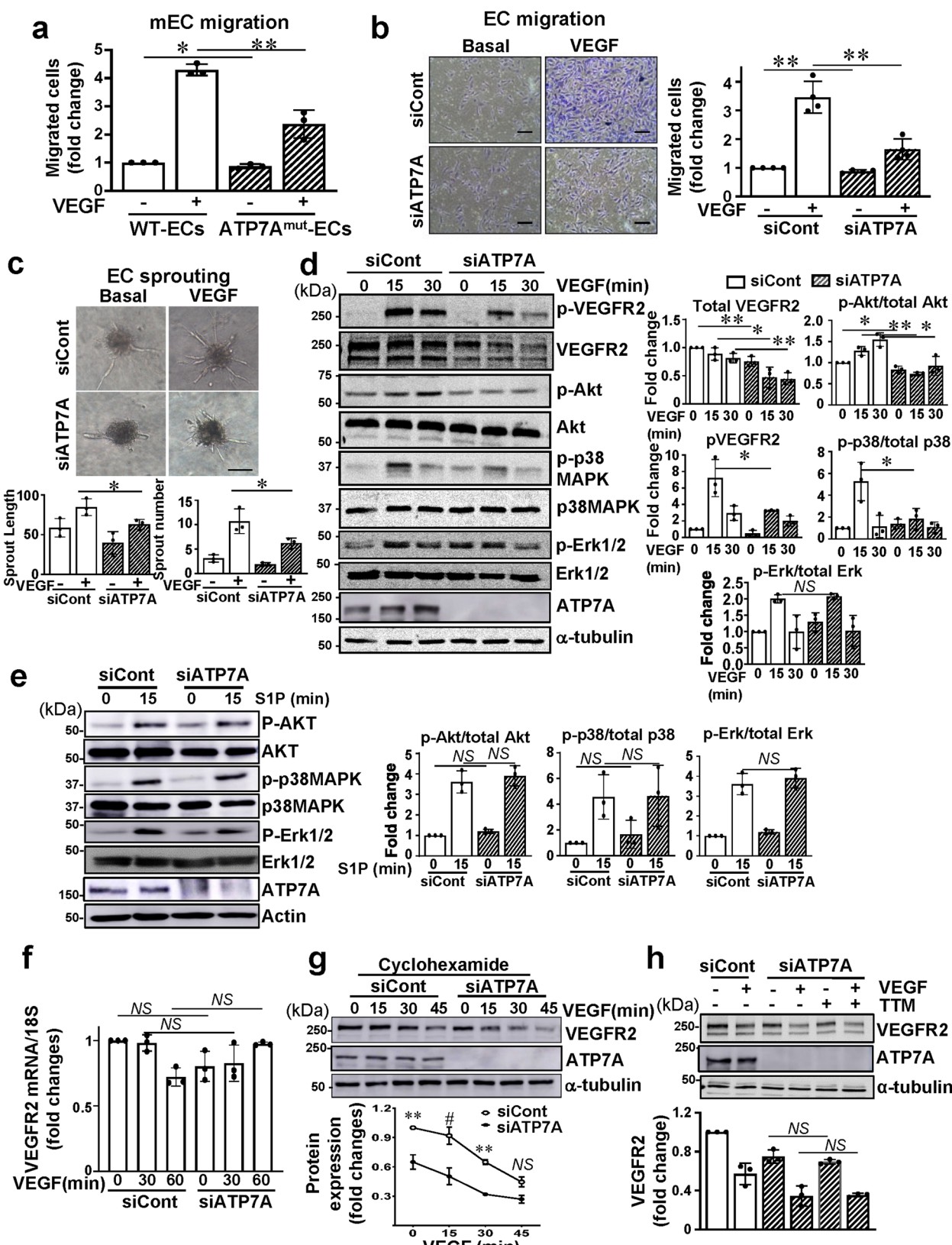

**ATP7A functions to limit lysosomal degradation of VEGFR2 in VEGF-stimulated ECs**. A cell surface biotinylation assay revealed that VEGF stimulation increased cell surface ATP7A levels within 5 min, followed by a gradual decrease over the next 15 min, which was also associated with a time-dependent reduction of cell surface VEGFR2, reflecting receptor (VEGFR2) internalization (Fig. 4a). Of note, ATP7A knockdown promoted VEGF-induced reduction of cell surface VEGFR2. Interestingly, Fig. 4b, c using co-localization analysis show that VEGF stimulation for 15 min promoted VEGFR2 internalization to Rab5-positive early endosomes, but not Lamp2-positive lysosomes, which was followed by trafficking to Rab11-positive recycling

**Fig. 2 ATP7A knockdown inhibits VEGFR2 signaling via promoting VEGFR2 degradation in a Cu-independent manner. a, b** Endothelial cell (EC) migration measured using modified Boyden chamber method in ECs isolated from Wild Type (WT) and ATP7A[mut] mice (**a**) or human umbilical vein endothelial cells (HUVECs) transfected with control or ATP7A siRNAs (**b**) stimulated with vascular endothelial growth factor (VEGF) (20 ng/ml) for 6 h. **a** $n = 3$, *$p = 0.0485$, **$p = 0.0034$; **b** $n = 3$, **$p = 0.0029$, **$p = 0.0016$ (two-tailed unpaired $t$-test). Scale bars = 100 μm. **c** EC spheroid spouting assay in HUVECs transfected with control or ATP7A siRNAs. Right panels show an average number of sprouts and tube length per field. Scale bars = 150 μm. $n = 3$, *$p = 0.038$, *$p = 0.0469$ (two-tailed unpaired $t$-test). **d, e** HUVECs transfected with control or ATP7A siRNAs were stimulated with VEGF (20 ng/ml) or sphingosine-1-phosphate (S1P) (10 μM) for the indicated time. Lysates were immunoblotted (IB) with indicated antibodies (Abs). α-tubulin as a loading control. The graph represents the averaged fold change over the basal control. **d** $n = 3$, VEGFR2 **$p = 0.0086$, *$p = 0.0272$, **$p = 0.0092$; pAKT/total AKT: *$p = 0.0279$, **$p = 0.0011$, *$p = 0.0177$; p-VEGFR2: *$p = 0.0393$; $p$-p38/total p38: *$p = 0.0413$; NS = non significant; **e**: $n = 3$, NS = non significant (two-tailed unpaired $t$-test). **f**. HUVECs transfected with control or ATP7A siRNAs were stimulated with VEGF (20 ng/ml) and mRNA level for VEGFR2 normalized by 18S was measured. $n = 3$, NS = non significant (two-tailed unpaired $t$-test). **g** HUVECs transfected with control or ATP7A siRNA pretreated with cyclohexamide (10 nM for 10 min) were stimulated with VEGF (20 ng/ml). Lysates were IB with anti-VEGFR2 or α-tubulin (loading control) Abs. $n = 3$, **$p < 0.01$, #$p < 0.0001$ (two-way ANOVA followed by Bonferroni's multiple comparison analysis). **h** HUVECs transfected with control or ATP7A siRNAs were pretreated with tetrathiomolybdate (TTM) (10 nM) for 24 h, followed by VEGF stimulation for 30 min. Lysates were IB with anti-VEGFR2 or α-tubulin Abs. $n = 3$, NS = non significant (two-tailed unpaired $t$-test). Data are mean ± SEM.

endosomes after 30 min. By contrast, in ATP7A-depleted ECs, VEGF stimulation for 15 min facilitated VEGFR2 internalization to Lamp2-positive lysosomes, which was followed by targeting Rab7-positive late endosomes involved in lysosomal degradation after 30 min. These results suggest that ATP7A functions to limit lysosomal degradation of VEGFR2 in VEGF-stimulated ECs. Furthermore, a lysosomal inhibitor (Chloroquine), but not proteasome inhibitors (Lactacystin and MG132), prevented ATP7A KD-induced VEGFR2 degradation (Fig. 4d), suggesting that VEGF-induced ATP7A binding to VEGFR2 may prevent VEGFR2 targeting to lysosomes for degradation. Since autophagy induces the lysosomal degradation of cellular proteins, we next examined the possible role of autophagy in ATP7A KD-induced VEGFR2 degradation. Figure 4e showed that bafilomycin A1, a vacuolar H + ATPase (V-ATPase) blocker which inhibits fusion between autophagosomes and lysosomes[39], prevented the VEGF-induced VEGFR2 degradation in ATP7A-depleted ECs. Furthermore, rapamycin, an autophagy inducer and inhibitor of mTOR[40,41,42], which increased LC3II expression, promoted reduction of VEGFR2 protein in both basal and VEGF-stimulated HUVECs (Supplementary Fig. 5). Thus, these results suggest that autophagy-dependent mechanisms are involved in the lysosomal degradation of VEGFR2 in ATP7A depleted ECs.

**ATP7A depletion induces autophagy and promotes VEGFR2 colocalization with light chain 3 (LC3).** To determine whether ATP7A depletion induces autophagy to promote lysosomal degradation of VEGFR2, HUVECs were transfected with a red fluorescent protein (RFP) and green fluorescent protein (GFP) double-tagged LC3 construct which shows the mature and immature LC3-positive puncta[43] (Fig. 5a). Since the GFP signal, but not the RFP signal, is sensitive to acidic conditions, immature autophagosomes show yellow dots due to intact GFP and RFP signals, while mature autophagolysosomes show only the RFP signal as the green GFP signal is acid-quenched. Figure 5a showed that RFP(+)GFP(−) mature autophagolysosome formation was slightly increased by ATP7A KD in the basal state, which was further enhanced after VEGF stimulation for 30 min in ECs. In order to examine if ATP7A regulates autophagy during angiogenesis in vivo, we used the hindlimb ischemia model and autophagy reporter CAG-RFP-EGFP-LC3 transgenic mice crossed with or without ATP7A[mut] mice. These autophagy reporter mice can detect newly formed autophagosomes with physiological pH expressing both RFP and EGFP yielding a yellow color, or mature autophagolysosome with acidification expressing RFP with reduced EGFP[44]. As shown in Fig. 5b, there was a significant increase in RFP (+) EGFP (−) LC3 dots in ischemic hindlimbs of ATP7A[mut] mice compared to WT mice. This was associated with

a significant increase in LC3 protein expression in ischemic tissues of ATP7A[mut] mice compared to WT mice (Fig. 5c). Taken together, these results show that the autophagy flux is increased in ATP7A-depleted ECs stimulated with VEGF or in ATP7A dysfunctional mice in response to ischemia in which angiogenesis is impaired. Transmission electron microscopy (TEM) further confirmed an increase in autophagosome and autophagolysosome formation in VEGF-stimulated ATP7A-depleted ECs compared to control ECs (Fig. 5d). Immunofluorescence analysis also demonstrated that ATP7A knockdown promoted the colocalization of autophagy receptor p62/SQSTM1 with LC3 (Fig. 5e), as well as VEGFR2 with LC3 (Fig. 5f) or Lamp2 (Fig. 4b) in HUVECs stimulated with VEGF. Thus, these results suggest that loss of ATP7A induces autophagy, thereby promoting autophagic and lysosomal degradation of VEGFR2.

**ATP7A protects VEGFR2 against degradation by preventing p62 binding to VEGFR2.** We then examined how VEGFR2 was targeted to the LC3-positive/Lamp2-positive autophagolysosomes in ATP7A-depleted ECs after VEGF stimulation. It has been shown that ligand-induced VEGFR2 ubiquitination facilitates lysosomal degradation of the receptor[45] and that p62/SQSTM1 brings ubiquitinated cargoes to autophagosome for degradation[19]. We thus examined the role of ATP7A in VEGFR2 ubiquitination and its interaction with p62/SQSTM1. Figure 6a shows that ATP7A depletion in ECs increased VEGFR2 ubiquitination in the basal state, which was further enhanced after VEGF stimulation. This was associated with a significant increase in VEGFR2 colocalization with p62/SQSTM1 in ATP7A-depleted HUVECs after VEGF stimulation (Fig. 6b). Consistent with this, co-immunoprecipitation assays showed that ATP7A knockdown increased VEGFR2 binding to p62/SQSTM1, which was associated with enhanced VEGF-induced VEGFR2 degradation (Fig. 6c). Significantly, ATP7A was not co-immunoprecipitated with p62/SQSTM1 in ECs with or without VEGF stimulation (Supplementary Fig. 6). To confirm further the specific role of ATP7A in preventing VEGFR2 binding with p62/SQSTM1, as well as VEGFR2 degradation, we performed rescue experiments by transfecting human myc-ATP7A plasmid in ATP7A-depleted bovine aortic endothelial cells (BAECs). Figure 6d shows that p62 binding to VEGFR2, as well as VEGFR2 downregulation in ATP7A KD ECs stimulated with VEGF, were inhibited by re-expression of myc-hATP7A. These results suggest that ATP7A binding to VEGFR2 may protect against VEGFR2 degradation by preventing p62 binding to VEGFR2.

It is known that the UBA domain of p62 interacts with ubiquitin and ubiquitin-tagged proteins and acts as a selective autophagy receptor to bring the marked proteins for

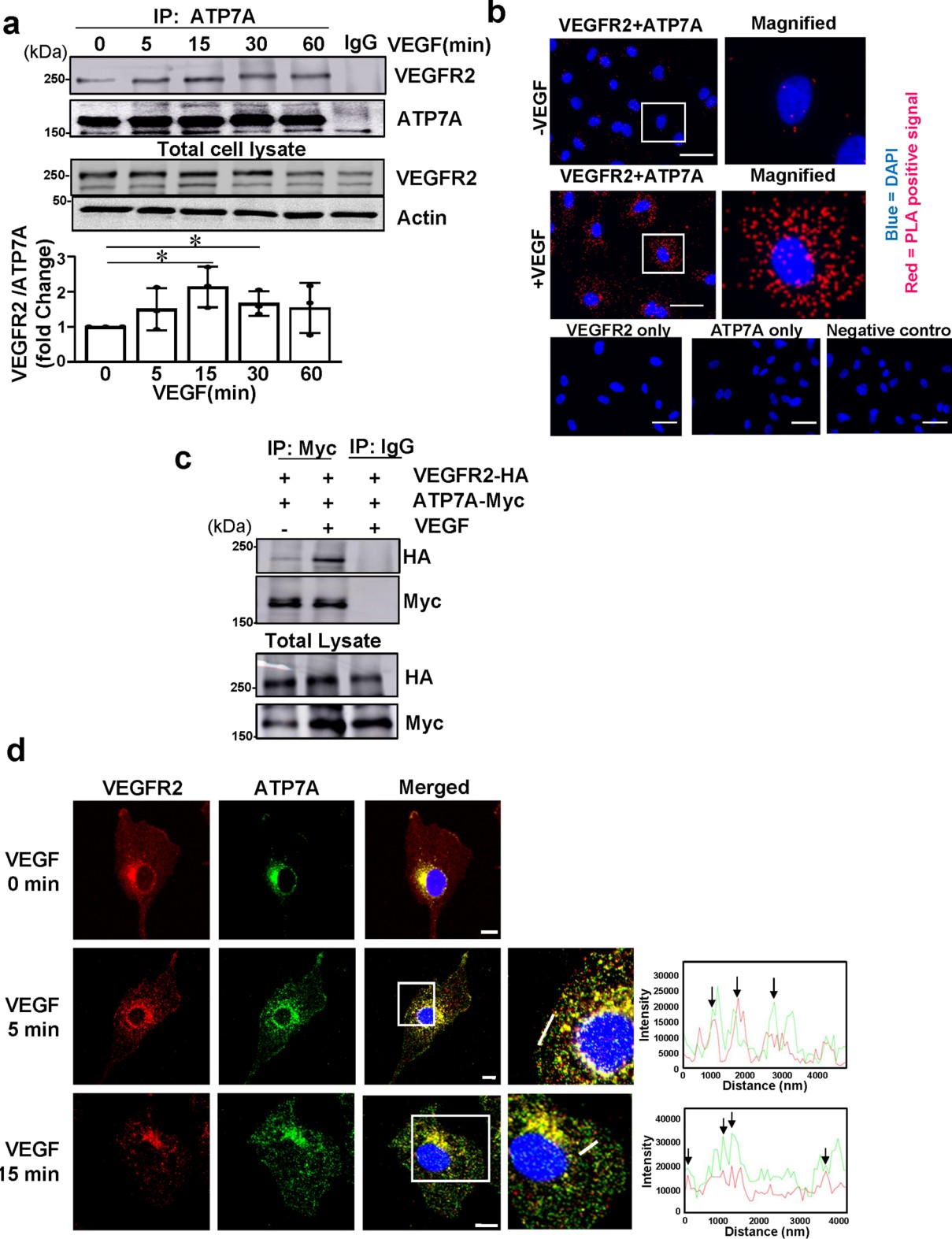

degradation[46]. We next examined the role of the p62 UBA domain in VEGFR2 degradation by transfecting HA-p62-WT or HA-p62 UBA domain deletion mutant (HA-p62 ΔUBA) in HUVECs. The data in Fig. 6e showed that overexpression of p62-WT, but not p62 ΔUBA, promoted VEGFR2 degradation in ECs with and without VEGF stimulation, as compared to vector control. Furthermore, overexpression of p62 significantly inhibited VEGF-induced EC migration (Fig. 6f). These results suggest that ATP7A protects against autophagic VEGFR2 degradation by preventing VEGFR2 ubiquitination and binding to p62/SQSTM1.

## Discussion

It has been shown that the canonical function of P-type ATPase transporter ATP7A is to deliver Cu to the secretary Cu-dependent enzymes such as LOX in the TGN and to export

**Fig. 3 VEGF stimulation promotes ATP7A binding with VEGFR2. a** Human umbilical vein endothelial cells (HUVECs) were stimulated with vascular endothelial growth factor (VEGF) (20 ng/ml) for the indicated time. Lysates were immunoprecipitated (IP) with anti-ATP7A antibody (Ab) followed by immunoblotting (IB) with VEGFR2 or ATP7A Ab. The graph represents the averaged fold change of the VEGFR2/ATP7A ratio over the basal ratio. $n = 3$, *$p = 0.0275$, *$p = 0.0303$ (two-tailed unpaired $t$-test). Data are mean ± SEM. **b** HUVECs stimulated with VEGF (20 ng/ml) for 30 min were fixed in 4% paraformaldehyde. In situ Proximity, Ligation Assay (PLA) was performed to show the interaction of ATP7A with VEGFR2. Red dots indicate PLA positive signal. Either ATP7A or VEGFR2 or no antibody was used as a negative control. The scale bar;10 μm. ($n = 3$). **c** Cos7 cells were transfected with ATP7A-myc and VEGFR2-HA with or without VEGF (20 ng/ml) stimulation. Lysates were subjected to anti-Myc IP and anti-HA IB. ($n = 3$). **d** Co-localization of VEGFR2 and ATP7A in HUVECs stimulated with VEGF (20 ng/ml), showing yellow fluorescence in the merged image, was analyzed by comparing the fluorescence intensity for each protein (white line on the enlarged image). Scale bars = 10 μm. ($n = 3$).

excess Cu from the cell[5,6]. This study provides evidence of crosstalk among Cu transporter ATP7A, the process of autophagy, and VEGFR2 degradation. We show that, in response to VEGF, ATP7A binds to VEGFR2 in ECs and functions to protect against p62/SQSTM1-mediated autophagic and lysosomal degradation of VEGFR2. In this way, ATP7A promotes VEGFR2 signaling, thereby driving angiogenesis (Fig. 7). Thus, endothelial ATP7A is identified as a potential therapeutic target to restore limb perfusion and revascularization in ischemic vascular disease via specific effects on VEGF receptor signaling.

This study demonstrates that ATP7A KD increased VEGFR2 ubiquitination and lysosomal degradation, thereby reducing VEGFR2 signaling. Of note, ATP7A KD had no significant effects on ligand-induced degradation of other tyrosine kinases such as IGFR1, FGFR1, and VEGFR3, or G-protein coupled receptor S1P-induced angiogenic signaling and EC migration in HUVEC. Mechanistically, experiments using co-immunoprecipitation, PLA, and co-transfection assays reveal that VEGF stimulation rapidly induces ATP7A binding to VEGFR2, which promotes VEGFR2 signaling and angiogenesis. It is shown that VEGF binding to VEGFR2 at the cell surface promotes receptor internalization to early endosomes, which is essential for activating sustained VEGFR2 signaling[2], while some fraction of the receptor is ubiquitinated and sorted to the lysosome for degradation[4] or else recycled to the plasma membrane[47]. By contrast, ATP7A is shown to be present predominantly at the TGN where it provides Cu to the secretory Cu enzymes, but in response to excess Cu, ATP7A translocates from the TGN to the plasma membrane via Rab22a to export Cu from the cells[10,48]. In this study, immunofluorescence analysis shows that VEGF promotes ATP7A translocation from the perinucleus to the plasma membrane where it colocalizes with VEGFR2 within 5 min, followed by their co-internalization to early endosomes where VEGFR2 signaling is propagated[49]. Consistent with our results, various non-metal stimulants such as insulin, NMDA, PDGF, and hypoxia[17,27,50,51] have been shown to induce re-localization of ATP7A from the TGN. These results suggest that ATP7A is specifically coupled to VEGFR2-mediated signaling and angiogenesis via binding to VEGFR2 in ECs.

The functional significance of endothelial ATP7A in angiogenesis in vivo is demonstrated utilizing tamoxifen-inducible EC-specific ATP7A deficient mice or ATP7A-dysfunctional ATP7A[mut] mice which showed impaired post-ischemic neovascularization. We found that ATP7A is highly expressed in CD31[+] angiogenic ECs in ischemic muscles. Bone marrow transplant experiments showed that ATP7A in tissue-resident cells, but not bone marrow cells, is required for ischemia-induced neovascularization. The process of angiogenesis includes EC migration and capillary tube formation. Consistent with this, ATP7A knockdown in HUVECs significantly inhibited VEGF-induced EC migration (using the modified Boyden chamber assay and wound scratch assay), as well as capillary tube formation using a spheroid EC sprouting assay. In addition, we showed that VEGF-induced EC migration is also inhibited in ECs isolated

from ATP7A[mut] mice compared to ECs from control mice. Thus, our data provide compelling evidence that ATP7A plays an essential role in VEGF-induced angiogenesis in ECs. The importance of ATP7A in endothelial function is also supported by our previous reports that ATP7A in VSMCs protects against endothelium-dependent vasodilation in hypertensive[14,52] or diabetic mice[16]. The role(s) of endothelial ATP7A in physiological angiogenesis, such as in embryonic and retinal angiogenesis, as well as in pathological angiogenesis (such as tumor angiogenesis and diabetic retinopathy) needs to be explored in the future studies.

Previous studies show that Ephrin B2[53], Nrp1[54], NUMB[55], or scaffold protein, IQGAP1[56] bind to VEGFR2 at the cell surface to promote VEGFR2 signaling and angiogenesis. Ephrin B2 is a transmembrane protein that promotes clathrin-dependent VEGFR2 internalization. Mutation of its C-terminal PDZ-binding motif impairs VEGFR2 endocytosis and hence angiogenesis[53]. The presence of a consensus PDZ-binding motif at the C terminal of ATP7A[57] suggests its functional similarity with Ephrin B2 in the regulation of VEGFR2 internalization. Nrp1 also binds to VEGFR2 in a ligand-independent manner and promotes VEGFR2 internalization through Rab11 and regulates angiogenesis[58]. However, we found that ATP7A KD promotes ligand-dependent VEGFR2 degradation without altering Nrp1 protein expression. We previously reported that PDGF stimulation in VSMCs promotes ATP7A binding to scaffold protein IQGAP1, which promotes Rac1-dependent VSMC migration and neointimal formation after vascular injury[59]. Thus, ATP7A may bind to the VEGFR2 along with its other binding partners, thereby promoting VEGFR2 signaling and angiogenesis. Addressing this complex binding scenario in detail is beyond the scope of this study, but will be the subject of future work.

It has been reported that VEGFR2 degradation is regulated by ubiquitin ligases βTRCP1[60] or by Cbl which polyubiquitylates the receptor[61] and that epsin promotes lysosomal degradation of ubiquitinated VEGFR2[62]. Thus, ATP7A binding to VEGFR2 may prevent receptor ubiquitination, which in turn may limit epsin-dependent VEGFR2 endocytosis, leading to lysosomal degradation[63]. Interestingly, to maintain homeostatic balance, newly synthesized VEGFR2 traffics through the Golgi to the plasma membrane to replenish the cell surface pool, in part via Golgi-localized t-SNARE family, syntaxin 6[64], KIF13B[65], or RNF121[66]. When this pathway is impaired, it facilitates lysosomal degradation of VEGFR2 resulting in inhibition of angiogenesis. Thus, ATP7A depletion at the TGN may prevent newly synthesized VEGFR2 trafficking through the Golgi to the cell surface, thereby facilitating lysosomal degradation. To support this notion, recent studies show that high glucose (as in diabetes) accelerates Golgi-localized VEGFR2 phosphorylation, which promotes VEGFR2 degradation, reducing replenishment of the surface pool[67].

Autophagy plays an important role in a wide range of physiological processes and human diseases. Kumar et al.[68] have reported that autophagy induction negatively regulates angiogenesis. In this

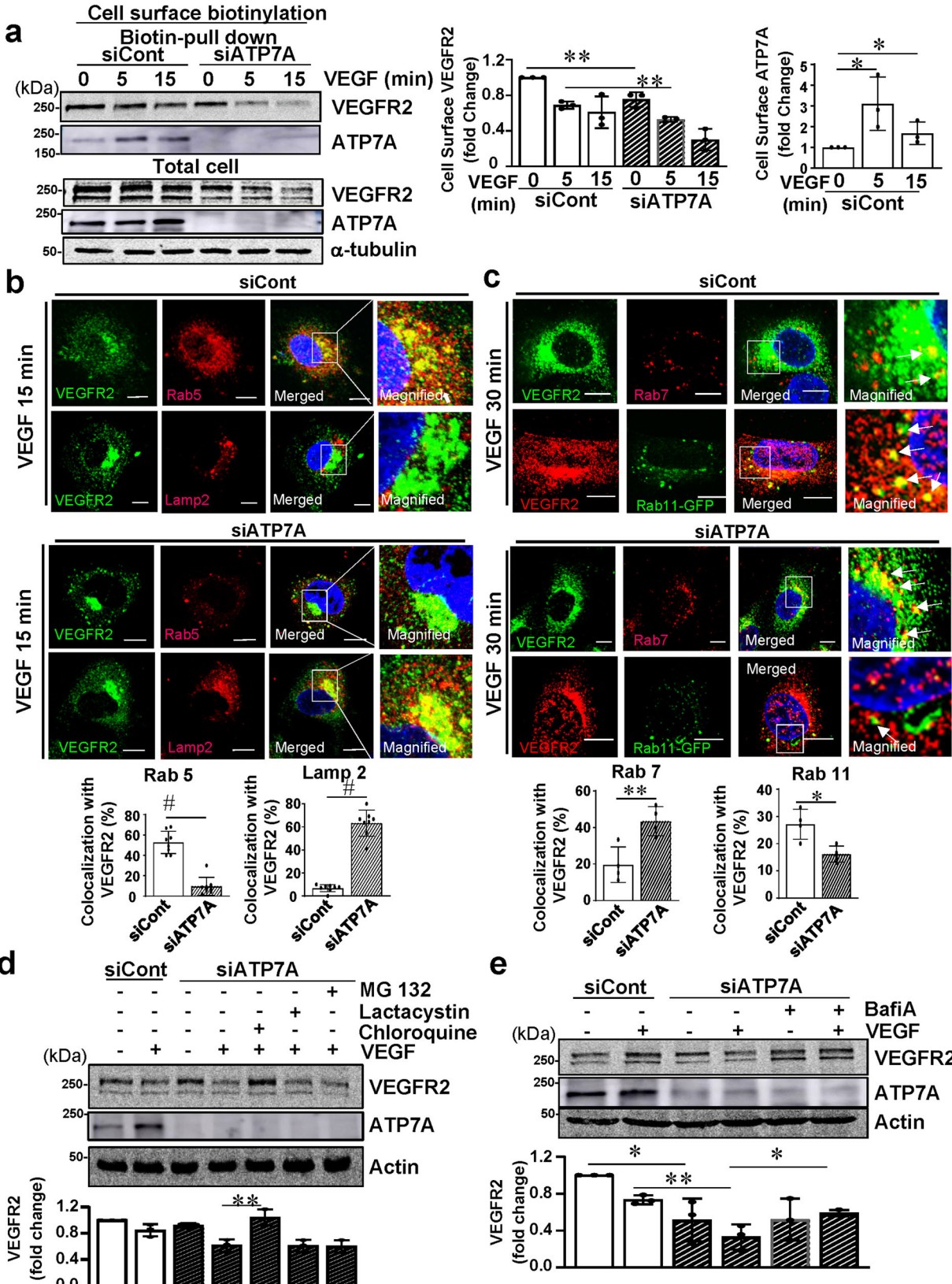

study, experiments using RFP-GFP-LC3 tandem reporter, cell surface biotinylation, immunofluorescence, and TEM, as well as inhibitors of lysosomal or proteasomal protein degradation or autophagy, and the autophagy inducer, rapamycin revealed that loss of ATP7A promotes autophagic flux, including autophagolysosome formation, thereby accelerating VEGFR2 protein degradation.

Furthermore, enhanced autophagy flux due to ATP7A dysfunction in vivo was supported by the autophagy reporter CAG-ATP7A$^{mut}$-RFP-EGFP-LC3 transgenic mice, which was associated with increased expression of autophagy proteins (LC3II), as well as increased RFP(+)EGFP (−)LC3 puncta in ischemic tissues of ATP7A$^{mut}$ mice. Thus, ATP7A dysfunction apparently induces

**Fig. 4 ATP7A knockdown enhances lysosomal degradation of VEGFR2. a** Human umbilical vein endothelial cells (HUVECs) transfected with control or ATP7A siRNAs were stimulated with vascular endothelial growth factor (VEGF) (20 ng/ml). Lysates were used for cell surface biotinylation using sulfo-NHS-SS-biotin to measure cell surface VEGFR2 or ATP7A by biotin pull-down, followed by immunoblotting (IB) with anti-VEGFR2 or ATP7A antibody (Ab). Non-immunoprecipitated (IP) total cell lysate was used for IB with Abs indicated. The bar graph represents the averaged cell-surface VEGFR2 or ATP7A protein levels over the basal control. $n = 3$, VEGFR2: $**p = 0.0066$, $**p = 0.0046$; ATP7A: $*p = 0.047$, $*p = 0.0343$ (two-tailed unpaired $t$-test). **b** HUVECs transfected with control or ATP7A siRNAs stimulated with VEGF (20 ng/ml) for 15 min were used for co-localization analysis for VEGFR2 and Rab5 or Lamp2. Yellow fluorescence in merged images indicates their colocalization. Scale bars = 10 μm. The bar graph represents the percentage of colocalization with VEGFR2. $n = 8$, Rab5: $\#p < 0.0001$; lamp2: $\#p < 0.0001$ (two-tailed unpaired $t$-test). **c** HUVECs transfected with control or ATP7A siRNAs and stimulated with VEGF (20 ng/ml) for 30 min were used for co-localization analysis for VEGFR2 and Rab7 using the corresponding antibodies. For VEGFR2-Rab11 colocalization, Rab11 was detected through Rab11-GFP plasmid over-expression and VEGFR2 was detected through an anti-VEGFR2 antibody. Yellow fluorescence in merged images indicates their colocalization. Scale bars = 10 μm. The bar graph represents the percentage of colocalization with VEGFR2. $n = 4$, Rab7: $**p = 0.009$; Rab11: $*p = 0.0128$ (two-tailed unpaired $t$-test). **d, e** HUVECs transfected with control or ATP7A siRNAs were pretreated with lysosome inhibitor chloroquine (500 μM for 30 min) or proteasome inhibitors MG132 (carbobenzoxy-Leu-Leu-leucinal) (10 μM for 30 min) for 30 min, or lactacystin (10 μM for 60 min) (**d**) or bafilomycin A1 (5 nM for 60 min) (**e**) followed by VEGF stimulation for 30 min. Lysates were IB with anti-VEGFR2 or actin loading control. **d** $n = 3$, $**p = 0.009$; **e** $n = 3$, $*p = 0.0223$, $**p = 0.0027$, $*p = 0.0122$ (two-tailed unpaired $t$-test). Data are mean ± SEM.

autophagy, thereby inhibiting post-ischemic neovascularization. Interestingly, it has been reported that the autophagic-lysosomal pathway is essential for ATP7A-mediated and ATP7B-mediated removal of excess Cu to protect against Cu-mediated toxicity in hepatocytes[23,24] or in senescent mice fibroblasts[25], indicating a linkage among autophagy, lysosomes, and Cu transport proteins for regulating Cu homeostasis. A recent report shows that Cu binds to autophagic kinases ULK1/2 to regulate their activity, leading to autophagosome complex formation and autophagic flux, which drives cell growth and survival in lung adenocarcinoma[69].

Our study suggests that ATP7A KD-induced autophagic VEGFR2 degradation is Cu-independent since the Cu chelator TTM is without effect on this response. It should be noted that the role of autophagy seems to differ, depending on the context of angiogenesis in physiological and pathological models, which is consistent with the notion that autophagy acts as a double-edged sword to regulate angiogenesis. For example, Sprott et al.[70] reported that EC-specific ATG5-deficient mice show reduced hypoxia/reoxygenation-triggered neovascularization in the retinopathy of prematurity model, suggesting that autophagy is required for pathological angiogenesis. On the other hand, it has been shown that autophagy impaired angiogenesis in vitro and in vivo[22]. We previously demonstrated that ATP7A protein expression is significantly reduced in type1 and type 2 diabetic vessels[16,17]. Thus, it is possible that ATP7A downregulation in diabetes may contribute to decreased VEGFR2 expression via inducing autophagy, leading to impaired angiogenesis in diabetic peripheral arterial disease. ATP7A downregulation may be an important issue in this pathology and will be the subject of future investigation.

The autophagic cargo/adapter p62/SQSTM1 is known as an adapter protein that delivers cargo and the substrates for autophagic degradation[71]. Recent studies have found that p62 interacts with ubiquitin and ubiquitin-tagged proteins via the UBA domain and acts as a selective autophagy receptor to delivers polyubiquitinated proteins to the autophagosome by interacting with the autophagosomal membrane protein, LC3. The sequestered cargo is subsequently degraded by lysosomal enzymes when the autophagosome fuses with a lysosome[72]. In this study, we show that loss of ATP7A increased VEGFR2 ubiquitination and recruitment of p62 to the VEGFR2 and their colocalization with LC3, leading to accelerated autophagic degradation of VEGFR2. The functional significance of p62/SQSTM1 is demonstrated by the result that overexpression of p62 reduced the VEGFR2 protein expression and VEGF-induced EC migration while p62 UBA domain deletion mutant had no effects on VEGFR2 expression (see Fig. 6e). Consistent with our

results, a previous study showed that p62 is involved in the starvation-induced autophagic degradation of IRS-1[73]. Of note, we found that VEGF stimulation promotes ATP7A binding to VEGFR2, but not to p62 in ECs (Fig. 3 and Supplementary Fig. 6). These observations suggest that ATP7A prevents interaction between the ubiquitinated VEGFR2 and the p62/SQSTM1 via binding to VEGFR2, thereby limiting the autophagic degradation of VEGFR2, which in turn promotes VEGF-induced angiogenesis in ECs. Given that p62 is a multifunctional scaffolding protein that interacts with various proteins to regulate diverse processes including apoptosis, as well as redox state via regulating the KEAP1-NRF2 pathway[74], other p62-mediated mechanisms may be involved in p62-induced impaired angiogenic responses. Further studies are necessary to understand the other mechanisms by which ATP7A dysfunction induces autophagy in ECs or other cell types.

The canonical role of ATP7A is to deliver Cu to the secretory Cu-dependent enzymes including SOD3 and LOX to regulate its activity[5,6]. We and others showed that LOX promotes tumorigenesis and metastasis[7] and is partially involved in VEGF-induced angiogenesis in ECs[8,9]. We reported that SOD3 is required for post-ischemic neovascularization[9]. Surprisingly, we found that the Cu chelator TTM[30–34,51,75] had no effect on either ATP7A depletion-induced VEGFR2 degradation of VEGFR2 signaling. Of note, the chelator efficacy of TTM was confirmed by showing that TTM almost completely inhibited VEGF-induced Cu-dependent LOX activity, as well as CuCl₂-induced phosphorylation of ERK, which is downstream of MEK which is directly activated by Cu[36,37]. These results suggest that the Cu-LOX axis-dependent ATP7A-mediated angiogenesis may be downstream of ATP7A-mediated Cu-independent regulation of VEGFR2 signaling. In previous studies, we demonstrated that ATP7A in VSMCs plays an important role in PDGF-induced vascular migration in part via delivering Cu to LOX[27,59]; however, a role for ATP7A in regulating PDGFR expression levels in VSMC was not investigated. In ECs, since ATP7A KD had no significant effects on ligand-induced degradation of other tyrosine kinases and S1P-induced angiogenic signaling and EC migration, ATP7A seems not to be involved in all angiogenic responses.

In summary, our study demonstrates the endothelial ATP7A-autophagy linkage in regulating VEGFR2 signaling and provides evidence that ATP7A functions to promote VEGF-induced angiogenesis via limiting autophagic VEGFR2 degradation, in addition to its canonical function in delivering Cu to secretory Cu enzymes. These findings provide insights into endothelial ATP7A as a potential therapeutic target for the treatment of ischemic cardiovascular diseases.

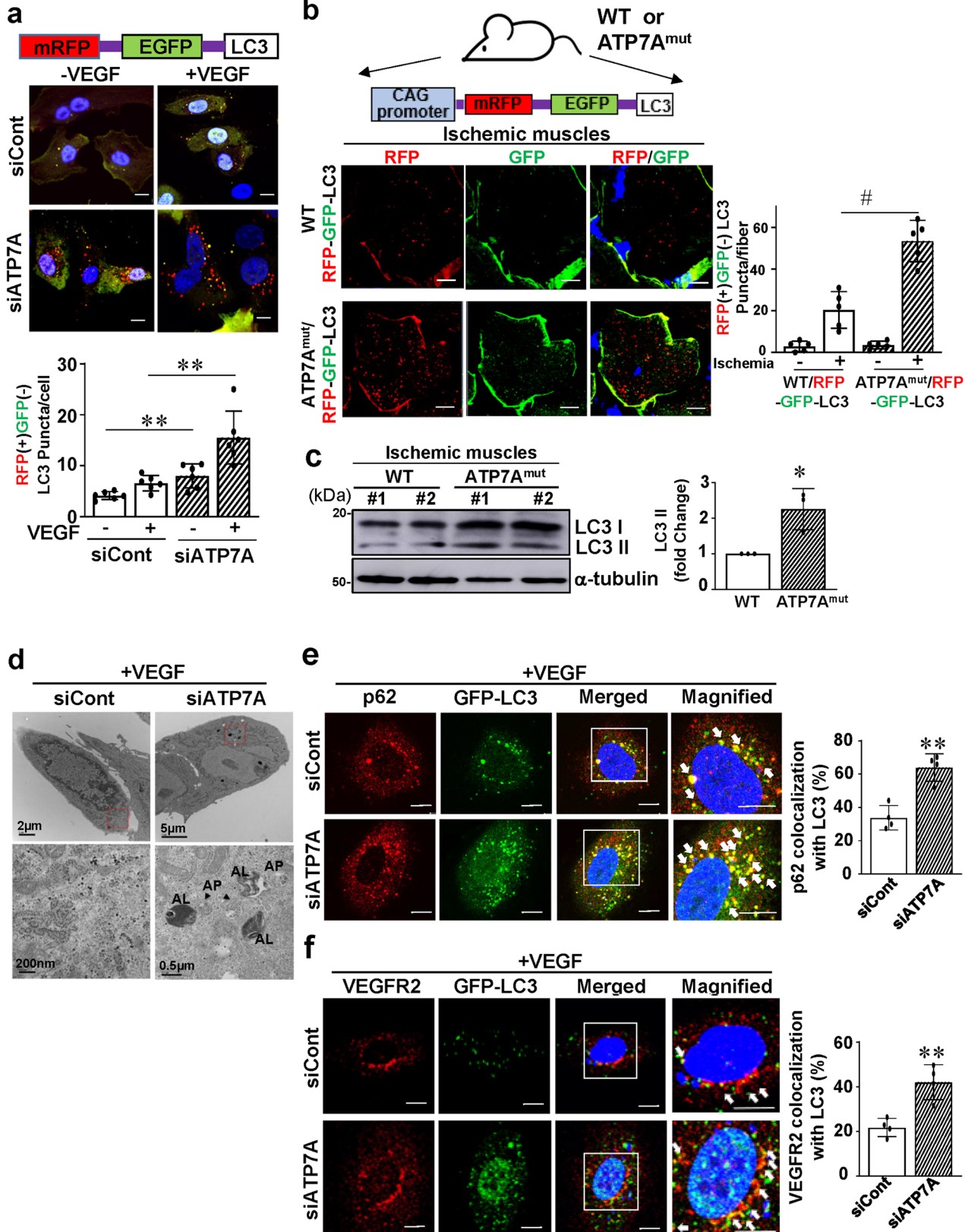

## Methods

**Animals**. The use of mice was in accordance with the National Institutes of Health Guide for the Care and Use of Laboratory Animals and relevant ethical regulations. Mouse experiments were approved by the Institutional Animal Care and Use Committee of Augusta University. To generate EC-specific ATP7A KO mice, we first generated an ATP7A floxed mouse line using ATP7A floxed ES cells obtained from European Conditional Mouse Mutagenesis Program (EUCOMM). We removed the NEO resistance gene by breeding with Flp recombinase expressing transgenic mouse. Inducible EC-specific hemizygous male ATP7A knockout (iEC-ATP7A KO) mice were generated by crossing homozygous floxed females (ATP7A$^{fl/fl}$) with mice expressing tamoxifen-inducible Cre recombinase under the control of the VE-cadherin promoter (ATP7A$^{+/Y}$;Cdh5-CreERT2$^{+/-}$). ATP7A deletion was initiated by daily intraperitoneal injections of 40 µg tamoxifen (Sigma-Aldrich)/g body weight for 10 days with 2 days break. Control mice, negative for

**Fig. 5 ATP7A depletion induces autophagy and promotes VEGFR2 colocalization with LC3. a** Immunofluorescence analysis of LC3 (Microtubule-associated protein 1 A/1B-light chain 3)-RFP and GFP puncta (mature and immature LC3 puncta, respectively), human umbilical vein endothelial cells (HUVECs) transfected with LC3-RFP-GFP plasmids in the presence of either ATP7A or control siRNAs were treated with vascular endothelial growth factor (VEGF) (20 ng/ml) for 30 min. Scale, 10 μm. The bar graph represents averaged number of RFP positive and GFP negative puncta per cell. $n = 6$, **$p = 0.0033$, **$p = 0.0022$ (two-tailed unpaired $t$-test). **b**. ATP7A$^{mut}$ mice crossed with RFP-EGFP-LC3 transgenic mice to generate ATP7A$^{mut}$/RFP-EGFP-LC3 transgenic reporter mice. Wild Type (WT) and ATP7A$^{mut}$ reporter mice were subjected to hind limb ischemia. Gastrocnemius muscles were harvested at 3 days after ischemia and examined for RFP and GFP expression. The scale bar; 10 μm. Bar graph represents averaged number of RFP positive and GFP negative puncta per fiber. $n = 5$, #$p = 0.0005$ (two-tailed unpaired $t$-test). **c** Representative western analysis of ischemic gastrocnemius muscles of WT and ATP7A$^{mut}$ mice at 3 days after ischemia. The bar graph represents averaged fold change over control and tubulin as a loading control. $n = 3$, *$p = 0.0196$ (two-tailed unpaired $t$-test). **d** HUVECs transfected with control or ATP7A siRNAs were stimulated with VEGF (20 ng/ml) for 30 min and were analyzed by transmission electron microscopy. Double membrane indicates autophagosome (AP) and black vesicle included AP indicated autolysosome (AL). ($n = 3$). **e, f** HUVECs transfected with LC3-GFP plasmid in the presence of either ATP7A or control siRNAs were stimulated with VEGF (20 ng/ml) for 30 min. Cells were stained with anti p62 or VEGFR2 antibodies. The right panels depict magnified images of the boxed areas seen in the left panels. The scale bar: 10 μm. Bar graph represents averaged number of p62 or VEGFR2 and LC3 colocalized puncta per cell. **e** $n = 4$, **$p = 0.0016$; **f** $n = 4$ **$p = 0.0036$ (two-tailed unpaired $t$-test). Data are mean ± SEM.

Cre expression received the same treatment. The mice were allowed to recover for 7 days after the last tamoxifen treatment before experiments. CAG-RFP-EGFP-LC3 transgenic mice expressing RFP and EGFP in a pH-dependent manner were bought from Jackson Laboratory (Stock No:027139). In some experiments, we used hemizygous male blotchy ATP7A mutant (ATP7A$^{mut}$) mice, which were generated by backcrossing ATP7A$^{mut}$ mice on C57BL/6JEiJ background (Jackson Laboratory (Bar Harbor, Maine)) with C57BL/6J mice for more than 10 generations[16]. Age-matched C57BL6 mice used for control, wild-type (WT) mice, were purchased from Jackson Laboratory. ATP7A$^{mut}$ mice carrying the X-linked blotchy ATP7A mutation have a splice site mutation introducing a new stop codon at amino acid residue 794 and show impaired copper transport function but survive to more than 6 months of age. All mice were maintained at the Augusta University animal facility. Room temperature and humidity were maintained at 22.5 °C and between 50% and 60%, respectively. All mice were held under the 12:12 (12-h light:12-h dark) light/dark cycle. Mice were held in individually ventilated caging with a maximum of 5 or a minimum of 2 mice per cage. Mice at 8 to 12 weeks old were used for experiments.

**Hindlimb ischemia model**. Mice were subjected to unilateral hindlimb surgery under anesthesia with intraperitoneal administration of ketamine (87 mg/kg) and xylazine (13 mg/kg). We performed ligation and segmental resection of left femoral vessels followed by physiological and histological analysis[76]. Briefly, the left femoral artery was exposed, ligated both proximally and distally using 6–0 silk sutures and the vessels between the ligatures were excised without damaging the femoral nerve. All arterial branches between the ligations were obliterated using an electricalcoagulator (Fine Scientific Tools). Skin closure was done using 6–0 nylon sutures. We measured ischemic (left)/nonischemic (right) limb blood flow ratio using a laser Doppler blood flow (LDBF) analyzer (PeriScan PIM 3 System; Perimed). Mice were anesthetized and placed on a heating plate at 37 °C for 10 min to minimize temperature variation. Before and after surgery, LDBF analysis was performed in the plantar sole. Blood flow was displayed as changes in the laser frequency, represented by different color pixels, and mean LDBF values were expressed as the ratio of ischemic to nonischemic LDBF.

**Bone marrow (BM) transplantation**. BMCs were isolated by density gradient separation. Recipient mice were lethally irradiated with 9.5 Gy and received an intravenous injection of 3 million donor bone marrow cells 24 h after irradiation. To determine the transplantation efficiency, transplantation was performed between WT mice (Jackson Laboratories) and either ATP7A$^{mut}$ or WT mice. Hindlimb ischemia was induced 6 to 8 weeks after bone marrow transplantation.

**Cell culture**. The primary HUVECs (Human Umbilical Vein Endothelial Cells) purchased from Lonza (CC-2519, USA) were cultured in EndoGRO (EMD Millipore) with 5% fetal bovine serum (FBS) (Atlanta Biological) until passages 6. Bovine aortic endothelial cells (BAEC; VEC Technologies) and COS-7 cells (ATCC® CRL-1651™) were grown in Dulbecco's modified Eagle's medium (DMEM) containing 10% (vol/vol) fetal bovine serum and used for experiments until passage 10.

**Histological analysis**. For cryosections, mice were euthanized and perfused through the left ventricle with saline and 4% paraformaldehyde, limbs were fixed in 4% paraformaldehyde (PFA) overnight and incubated with 30% sucrose, and adductor and gastrocnemius muscles were embedded in OCT compound (Sakura Finetek). For paraffin sections, we performed methanol fixation or PFA fixation with decalcification by Immunocal (Decal Chemical Corp.). Capillary density in the ischemic muscles was determined in 7 μm cryosections or in 5 μm methanol fixed

paraffin sections that were stained with anti-mouse CD31 antibody (BD)(1:300). For immunohistochemistry, we used R.T.U. Vectorstain Elite (Vector Laboratories) followed by DAB visualization (Vector Laboratories). Images were captured by Axio scope microscope (Zeiss) or confocal microscopy (Zeiss) and processed by AxioVision 4.8 or LSM510 or ZEN 2.3 software (Zeiss).

**Immunofluorescence analysis**. Immunofluorescence staining was performed in fresh frozen OCT tissue sections with primary antibodies against ATP7A (LS Bioscience)(1:300) and CD31 (BD Bioscience) (1:300). Secondary antibodies were Alexa Fluor 488 or 546-conjugated goat anti-mouse IgG and goat anti-rat IgG (Invitrogen)(1:500). Tissue sections were mounted using Vectashield (Vector Laboratories) containing DAPI for nuclear counter-staining. For cultured cells, HUVECs on glass coverslips were rinsed quickly in ice-cold PBS, fixed in freshly prepared 4% paraformaldehyde in PBS for 10 min at room temperature, permeabilized in 0.05% Triton X-100 in PBS for 5 min, and rinsed sequentially in PBS, 50 μmol/L NH$_4$Cl and PBS for 10 min each. After incubation for 1 h in blocking buffer (PBS + 3%BSA), cells were incubated with primary antibody for 18 h at 4 °C, rinsed in PBS/BSA, and then incubated in Alexa Fluor 488-conjugated IgG for 1 h at room temperature, and cells rinsed with PBS. Cells on coverslips were mounted onto glass slides using Vectashield (Vector Laboratories) and observed using confocal microscopy.

**Immunoprecipitation and Immunoblotting**. Cells were lysed in lysis buffer, pH 7.4 (in mM) 50 HEPES, 5 EDTA, 120 NaCl), 1% Triton X-100, protease inhibitors (10 μg/ml aprotinin, 1 mmol/L phenylmethylsulfonyl fluoride, 10 μg/ml leupeptin) and phosphatase inhibitors (mmol/L) 50 sodium fluoride, 1 sodium orthovanadate, 10 sodium pyrophosphate). For immunoprecipitation, cell lysates (1,000 μg) were precipitated with antibody overnight at 4 °C and then incubated with 20 μl of protein A/G-agarose beads for 1.5 h at 4 °C. Cell lysates (25 μg) or immunoprecipitates were separated using SDS-polyacrylamide gel electrophoresis. Goat anti-VEGFR2 (R&D System), Mouse anti-p62 (BD), and Mouse anti-α-tubulin (Santa Cruz) were incubated on nitrocellulose membranes containing protein with 1:1000 dilution. For protein expression in ischemic muscle, mice were perfused with cold phosphate buffer saline. Muscle samples were harvested and frozen in liquid nitrogen. Muscle samples were crushed and lysed with RIPA lysis buffer (5 mM Tris-HCl (pH 7.6), 150 mM NaCl, 1% NP-40, 1%sodium deoxycholate, 0.1% SDS) with protease inhibitor followed by brief sonication. An equal amount of protein was separated by SDS-PAGE. Following primary antibodies were used: anti-ATP7A (LS Bioscience) and anti-LC3 (MBL) at 1:1000 dilution. The other antibodies used in this study are anti-VEGFR3 (1:1000) from the R&D System, anti-AKT1/2/3 (1:1000), anti-Flk1(A3) (1:1000), anti-ubiquitin P4D1 (1:1000) from Santa Cruz, anti-p-p44/42 MAPK (1:1000), anti-p-44/42 MAPK (1:1000) from Cell Signaling and Goat Anti-Rabbit IgG-HRP conjugate (1:2000), Goat Anti-mouse IgG-HRP conjugate (1:2000) from Bio-Rad. Protein expression was visualized by ECL (Amersham). Band density was quantified by ImageJ.

**Transfections**. Control siRNA and ATP7A siRNA were obtained from Applied Biosystems and Sigma Aldrich. Sequence of ATP7A siRNA was as follows: sense, 5′-CUGUAUUAGUAGCAGUUGA-3′; antisense, 5′-UUCUGAUAAAGAUCGG CUGCA-3′. HUVECs were seeded into culture dishes 1 day before transfection. Transfection of siRNA (30 nM) was performed using Oligofectamine (Invitrogen) according to the manufacturer's protocol. VEGF stimulation was performed 48 h after transfection. For double transfection of siRNA and plasmid DNA, cells were transfected with siRNAs (20 nM) and plasmid DNA (3 μg for 100 mm dishes) using JetPRIME (Polyplus, USA). For mRFP-GFP-LC3 plasmid which was a gift from Tamotsu Yoshimori (Addgene plasmid # 21074; http://n2t.net/addgene:21074; RRID: Addgene_21074), cells were transfected with DNA (1–3 μg) using polyethylenimine (PEI, Polysciences, USA). After transfection, cells were

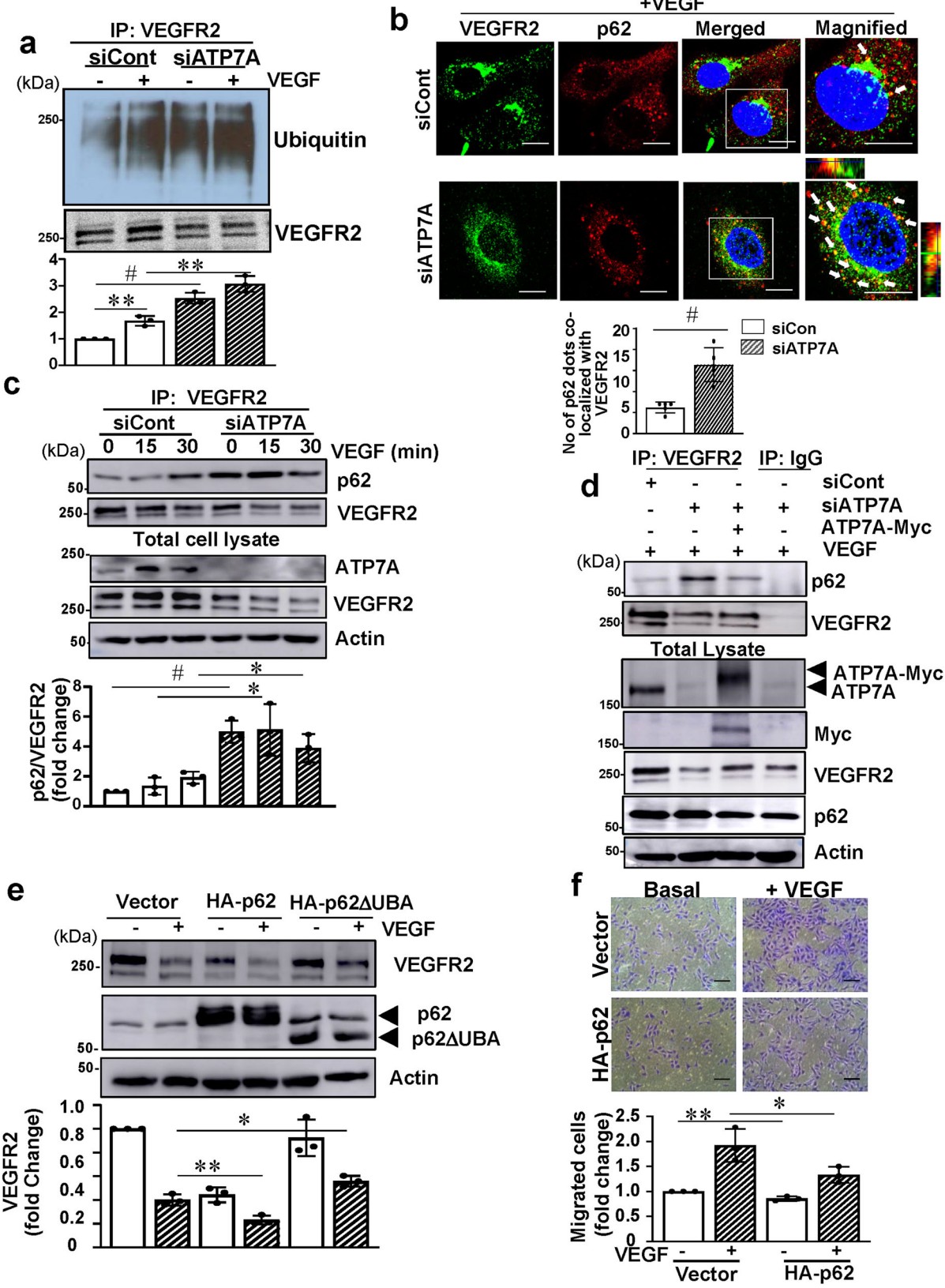

changed to a growth medium and further incubated for 48 h at 37 °C before experiments.

**Transmission electron microscopy (TEM)**. HUVECs transfected with siCont or siATP7A were washed with Sorenson buffer (pH 7.4) and fixed with 2% glutaraldehyde in Sorenson buffer (pH 7.4) for 30 min on 4 °C. The images were taken by

JEM-1220 (JEOL) at the electron microscopy core facility at the University of Illinois at Chicago.

**Modified Boyden chamber assay**. HUVEC were plated at $1 \times 10^5$ cell suspension/chamber in low serum media with 10 mM dimedone or the vehicle onto the upper chamber of Transwell inserts containing 8 μm pore (BD-Biosciences). Chemotaxis

**Fig. 6 ATP7A protects against VEGFR2 degradation by preventing VEGFR2 ubiquitination and its binding with p62. a** Human umbilical vein endothelial cells (HUVECs) transfected with control or ATP7A siRNAs were stimulated with vascular endothelial growth factor (VEGF) (20 ng/ml) for 30 min in the presence of N-ethylmaleimide (5 mM). Lysates were IP with VEGFR2 followed by immunoblotting (IB) with anti Ubiquitin antibody (Ab) to detect VEGFR2 ubiquitination. The bar graph represents averaged fold change over the control. $n = 3$, $**p = 0.0031$, $\#p = 0.0002$, $**p = 0.0023$ (two-tailed unpaired $t$-test). **b** HUVECs transfected with ATP7A or control siRNAs were stimulated with VEGF (20 ng/ml) for 30 min and analyzed by immunofluorescence using VEGFR2 and p62 Abs. The scale bar = 10 μm. Bar graph represents averaged number of VEGFR2 and p62 colocalized dots per cell. $n = 5$, $\#p = 0.0007$ (two-tailed unpaired $t$-test). **c** HUVECs transfected with control or ATP7A siRNAs were stimulated with VEGF (20 ng/ml) for 15 and 30 min. Lysates were immunoprecipitated (IP) with VEGFR2 Ab followed by IB with p62 or VEGFR2 Abs. $n = 3$ independent experiments, $\#p = 0.0007$, $*p = 0.0232$, $*p = 0.0294$ (two-tailed unpaired $t$-test). **d** Bovine aortic endothelial cells (BAEC) transfected with bovine control or ATP7A siRNAs along with empty vector or ATP7A-Myc plasmid and cells were stimulated with VEGF (20 ng/ml) for 30 min. Lysates were IP with VEGFR2 Ab followed by IB with p62 or VEGFR2 Abs. ($n = 3$). **e** HUVECs transfected with Adenovirus expressing p62 WT or p62 lacking ubiquitin-associated (UBA) domain were treated with VEGF (20 ng/ml) for 60 min. Lysates were used for IB with VEGFR2, HA-p62. $n = 3$ independent experiments,$*p = 0.0145$, $**p = 0.0082$ (two-tailed unpaired $t$-test). **f** HUVECs transfected with empty vector or HA-p62 were used to measure basal and VEGF-induced endothelial cell (EC) migration using the modified Boyden chamber method. Bar graph represents averaged fold change over control. Scale bars = 100 μm. $n = 3$, $**p = 0.0059$, $*p = 0.0499$ (two-tailed unpaired $t$-test). Data are mean ± SEM.

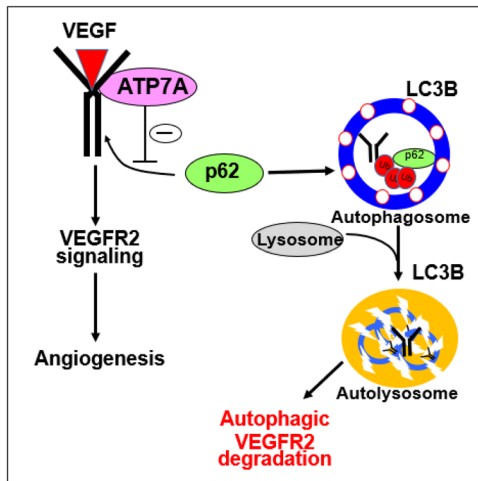

**Fig. 7 Proposed model.** Proposed model showing that ATP7A binds to VEGFR2 in endothelial cells in response to VEGF, which protects against p62/SQSTM1-mediated autophagic and lysosomal degradation of VEGFR2. This in turn promotes VEGFR2 signaling, thereby promoting angiogenesis and ischemic neovascularization.

was achieved by the presence of 50 ng/ml VEGF with 10 mM dimedone or vehicle in a 37 °C culture incubator. After 6 h, transwell chambers were rinsed with PBS, and the cells on top were removed with a cotton tip applicator. Cells on the bottom of the transwell were fixed in 100% methanol and stained with Diff-Quick Stain kit from Imed Inc. Eight random fields per transwell were imaged with a Nikon digital camera, and the number of cells was counted by using NIH Image J software.

**3D spheroid sprouting assay**. To generate HUVEC spheroids, the endothelial cells were cultured overnight in a complete medium containing 0.25% carboxyl methylcellulose (CMC) in non-adherent U bottom 96 well plates. Harvested spheroids were overlaid with 0.8% CMC and 20% FBS solution in M199 media and collagen solution added in a 1:1 v/v ratio and plated on pre wormed 24 well plates to form the gel. The gels were overlaid with a complete medium supplemented with 50 ng/ml VEGF. Twenty-four hours after embedding the cumulative length of the sprouts growing out of each spheroid, their mean number and length were measured using ImageJ software on bright-field images.

**Cell surface biotinylation assay**. The cell surface VEGFR2 or ATP7A was biotinylated following exposure to VEGF (20 ng/ml) for the indicated time. Biotinylation of cells was carried out using the cell-impermeable, thiol-cleavable Sulfo-NHS-SS-biotin (Pierce) reagent to label cell surface proteins. All of the biotinylation procedures were carried out at 4 °C. Briefly, cell-surface proteins were labeled for 45 min with 0.2 mg/ml cleavable water-soluble cell-impermeable sulpho-NHS-SS-biotin (Pierce Endogen) in PBS supplemented with 0.5 mM MgCl2 and 1 mM CaCl2. The unbound sulfo-NHS-SS-biotin was quenched by 50 mM Tris (pH 8.0) containing 100 mM NaCl. The cells were lysed in 500 μl of ice-cold lysis buffer, pH 7.4 (50 mM HEPES, 5 mM EDTA, 100 mM NaCl), 1% Triton X-100, 60 mM n-Octyl-β-D-glucopyranoside, protease inhibitors (10 μg/ml aprotinin, 1 mM

phenylmethylsulfonyl fluoride, 10 μg/ml leupeptin) and phosphatase inhibitors (50 mM sodium fluoride, 1 mM sodium orthovanadate, 10 mM sodium pyrophosphate), and precipitated on streptavidin–agarose beads (Pierce Endogen) overnight. The beads were washed three times in lysis buffer and were eluted in loading buffer. Proteins were subjected to Western immunoblot analysis to detect biotinylated VEGFR2 or ATP7A.

**Isolation of mouse aortic endothelial cells (MAECs)**. Primary MAECs were isolated as follows[77]. Mice were anesthetized and perfused with PBS containing 1000 U/mL of heparin. The aorta was removed from the mouse and washed with ice-cold PBS, and the surrounding fat tissues were removed. The aorta was then immediately transferred to endothelial growth medium and cut into 2 mm small segments. The collagen gel was prepared by diluting type I collagen with an endothelial growth medium. The collagen gel was added to 6-well plates (1.5 ml per well) and allowed to solidify at 37 °C for at least 30 min. The aorta was cut open and carefully laid on a collagen gel with the endothelium directly facing the gel. The aorta explants were incubated at 37 °C in an endothelial growth medium. The aorta fragments were removed after 2 days. The remaining adherent endothelial cells were allowed to grow on the matrix for additional 3 days. Thereafter, the cells were harvested by 0.25% Trypsin solution and centrifuged for 5 min, and placed onto a 0.1% gelatin-coated flask.

**Proximity ligation assay (PLA)**. An in situ proximity ligation assay (PLA) was performed by using Duolink In Situ PLA reagents (Sigma-Aldrich) to assess the endogenous interaction of ATP7A and VEGFR2 with high specificity and sensitivity. Cells were fixed with 4% paraformaldehyde for 10 min, and then permeabilized in 0.05% Triton X-100 in PBS for 5 min and incubated with primary antibodies against ATP7A (mouse monoclonal, 1:200, Life Span Biosciences, Cat# LS-B8162)) and VEGFR2 (rabbit monoclonal, 1:200, Cell Signaling, Cat# 2479 S) for overnight at 4 °C. After washing, cells were treated with secondary anti-mouse and anti-rabbit antibodies conjugated with oligonucleotides of a PLA probe and then subjected to oligonucleotide hybridization, ligation, amplification, and detection following the manufacturer's instructions. In this assay, a positive signal is created only when the epitopes of the target proteins are in close proximity (<40 nm). Finally, the signal from each detected pair of PLA probes in the cells with a mounting medium containing DAPI was then counted under Zeiss Confocal LSM 510 microscope ($\lambda_{ex}$ 594 nm; $\lambda_{em}$ 624 nm). For negative control, cells were treated with single or no primary antibody.

**LOX activity assay**. LOX activity from cell lysate was measured using a high-sensitivity fluorescence assay[78]. Cells were homogenized in 1× LOX Urea buffer and protein estimation was done using a standard method. An equal amount of protein samples was incubated in the presence and absence of 500 μM TTM at 37 °C for 30 min with a final reaction mixture supplied by Amplite Fluorimetric Lysyl Oxidase Assay kit (AAT Bioquest) to the manufacturer's instruction. The reaction was stopped on ice, and differences in fluorescence intensity (540-nm excitation wavelength and 590-nm emission wavelength) between samples with and without TTM were determined. Specific activity was determined by the ratio of activity to the relative amount of protein.

**Quantitative real-time PCR**. Total RNA was isolated by using Tri Reagent (Molecular Research Center, Inc.) and phenol/chloroform. Reverse transcription was carried out using a high-capacity cDNA reverse transcription kit (Applied Biosystems) with 2 μg of total RNA. The PCR was performed according to the manufacturer's protocol using ABI PRISM 7000 Sequence Detection System 26 (Applied Biosystems, CA) and QuantiFast SYBR Green PCR Kit (Qiagen, Valenica, Foster City, CA). Primer sequences for qPCR are listed in

Supplementary Table 1. Samples were all run in triplicates to reduce variability. The expression of genes was normalized and expressed as fold changes relative to HPRT or18S.

**Statistical analysis**. Data are presented as mean ± SEM. Statistical tests were performed using Prism v4 (GraphPad Software, San Diego, CA). Data were compared between groups of cells and animals by a 2-tailed unpaired Student $t$-test. Two-way ANOVA was applied for multiple comparisons, followed by Bonferroni's multiple comparison analysis. The statistical analysis was based on $n$, indicating the number of independent biological samples or mice as described in detail in the respective figure legends. Values of $*p < 0.05$, $**p < 0.01$, $\#p < 0.001$ were considered statistically significant.

**Reporting summary**. Further information on research design is available in the Nature Research Reporting Summary linked to this article.

## Data availability
Data underlying all figures and supplementary figures, uncropped and unprocessed immunoblot scans, and PCR agarose gel pictures for all figures and supplementary figures, are provided as Source Data. All other relevant data are available from the corresponding authors on request. Source data are provided with this paper.

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

## Acknowledgements

This work was supported by NIHR01HL135584 (to M.U.-F.), NIHR01HL133613, NIHR01HL116976 (to T.F., M.U.-F.), NIHR01HL070187 (to T.F.), Department of Veterans Affairs Merit Review grant 2I01BX001232 (to T.F.), and 17POST33660754 (to D.A.), Foundation Jerome Lejeune (to M.N.O., J.P.O.), NIHR01HL090651 (to J.P.O.), Department of Defense PR080428 (to J.P.O.), St.Baldrick's Foundation (to J.P.O.), Chicago Biomedical Consortium with support from the Searle Funds at The Chicago Community Trust (to J.P.O.), Center for Clinical and Translational Science at UIC (to J.P.O.), NIH National Center for Advancing Translational Sciences UL1TR00050 (to J.P.O.).

## Author contributions

M.U.-F., T.F., D.A., V.S., S.W.-Y. designed the study; D.A., V.S., S.W.-Y., M.N.O., J.P.O., A.D., M.M., Y.H. performed research; M.U.-F., T.F., D.A., V.S., S.W.-Y. analyzed data; J.P.O., J.H.K. discussed data and provided inputs; M.U.-F., T.F., D.A., V.S. wrote the manuscript; M.N.O., J.P.O., J.H.K. edited the manuscript. All the authors reviewed the manuscript.

## Competing interests

The authors declare no competing interests.
