## [Peer Review File · Nature Communications]

Reviewers' comments:

Reviewer #1 (Remarks to the Author):

This is an elegant study that adds important information to our understanding of VEGFR2 trafficking and signaling. The authors show that an endothelial-specific ATPase transporter ATP7A regulates VEGFR2 degradation.

The study is fairly complete but certain issues need to be addressed:

1. How specific is the ATP7A-VEGFR2 interaction? Does ATP7A ko (OR KD in vitro) affect degradation of other transmembrane proteins and in, particular receptor tyrosine kinases that are trafficked in a manner similar to VEGFR2 (e.g. IGF1R, Insulin receptor, etc.). and in manner distinct from VEGFR2 (e.g. FGFR1)?
2. Are there any effects of ATP7A KO on VEGFR3? Other VEGFR2 binding partners such as Nrp1?
3. It is important to fully investigate all aspects of VEGFR2 trafficking. While Rab5 and Lamp2 data are convincing it would be good to provide data regarding the col-localization extent with Rab11 and Rab7.
4. In Discussion it is important to put this new mechanism in the context of what is already known about VEGFR2 endocytosis and its control- in particular effects of Ephrin B2 and Neuropilin-1

Reviewer #2 (Remarks to the Author):

This is a study that evaluates the role of ATP7A in angiogenesis via VEGFR2. The topic builds on the author's previous studies of ATP7A and angiogenesis and it is unclear how novel the current results are compared to previous work. I find shortcomings in the experimental parts aiming to reveal the role of ATP7A. For example, when using siRNA, the authors should check how much the protein expression is downregulated in the cells by this treatment. We have found that this can vary a lot and optimization in terms of both conc of RNA and time of incubation is needed. The authors also claim a direct interaction between VEGFR2 and ATP7A using co-IP. The data in Fig 3A is not convincing. Truly there are no controls and it could be nonspecific binding or a tiny fraction of no biological relevance. This needs to be further addressed. Co-IP experiments are very complicated and depends on many parameters. To claim protein-protein interaction, additional experiments are needed. This is the one key of the paper, as the authors claim to reveal mechanistic information in this study..

Also, the authors state that the effects/pathway is independent of Cu. This is suggested based on TTM experiments, a chelator that would coordinate Cu. However, it is not shown that TTM enters the cells nor that it chelates Cu in the cell such that it is not available to ATP7A. Assessing Cu-dependence is hard as Cu is involved in so many things in cells and modulating the Cu levels will necessarily affect also other cellular processes. This should be further investigated before a claim can be made. Also, adding excess Cu should be included, and analysis (ICPMS) of internal or cell-associated Cu at different conditions determined to show that treatments have the expected effect. TTM is controversial as it can also bind to metal-binding domains in ATP7A via the Cu (forming a ternary complex).

Most of the experimental data differences have low significance, nothing better than one star and many changes noted are less than 2-fold. this is not highly convincing.

also, for co-localization between ATP7A and VEGFR2, the performed experiments cannot determine true co-localization as in near each other such that interaction takes place. the resolution is too low. truly, this only shows that the two proteins are in the same area of cells.

Also, likely these cells express ATP7B and parallel path involving ATP7B should be addressed (or excluded). ATP7B is supposed to mostly be in hepatic cells, but when one investigates most cells express some ATP7B.

Also, the antibody towards ATP7A needs to be verified as there are several and all do not work well.

Taken together, it is hard to judge the significance of the message, but regardless, I find many shortcomings in the experimental approach that should be addressed.

Reviewer #3 (Remarks to the Author):

This paper is obviously innovative. The main role of ATG7A originally is to maintain the copper homeostasis in cells. Copper ions are known to promote angiogenesis. This research found that ATP7A binds to VEGFR2 in ECs in response to VEGF and functions to protect against p62/SQSTM1 mediated autophagic and lysosomal degradation of VEGFR2 and promote VEGFR2 signaling in a Cu-independent manner, thereby promoting angiogenesis. The logic of this paper is clear.

However, the paper has several disadvantages:

(1) The paper proves that ATG7A can affect angiogenesis. But the authors do not prove that VEGF-induced EC angiogenesis in HUVECs transfected with ATP7A siRNA. Demonstrating angiogenesis in vitro could make the study more adequate.

(2) It is known that the UBA domain of p62 interacts with ubiquitin and ubiquitin-labeled proteins and acts as a selective autophagy receptor, degrading the labeled proteins. This paper proved that ATG7A inhibits the ubiquitination of VEGFR, and p62 can promote the ubiquitination of VEGFR through UBA, and ATG7A can inhibit the interaction of p62 and VEGFR, but there is no proof on how ATG7A works through p62-UBA. If the problem is proved, the research content will be more in-depth.

(3) The pictures of Western blot are not clear, such as Fig2.D, Fig5.C;

(4) The figures are not arranged neatly

We thank the reviewers for their careful reading of the manuscript and their constructive suggestions. We have carried out further experiments in the light of their questions. We have revised the manuscript in accordance with the reviewer's comments. The point-by-point responses to each of the reviewer's comments and questions are provided below.

Response to Reviewer # 1

Comments:

This is an elegant study that adds important information to our understanding of VEGFR2 trafficking and signaling. The authors show that an endothelial-specific ATPase transporter ATP7A regulates VEGFR2 degradation. The study is fairly complete but certain issues need to be addressed:

- 1. How specific is the ATP7A-VEGFR2 interaction? Does ATP7A ko (OR KD in vitro) affect degradation of other transmembrane proteins and in, particular receptor tyrosine kinases that are trafficked in a manner similar to VEGFR2 (e.g. IGF1R, Insulin receptor, etc.). and in manner distinct from VEGFR2 (e.g. FGFR1)?*
- 2. Are there any effects of ATP7A KO on VEGFR3? Other VEGFR2 binding partners such as Nrp1?*
- 3. It is important to fully investigate all aspects of VEGFR2 trafficking. While Rab5 and Lamp2 data are convincing it would be good to provide data regarding the col-localization extent with Rab11 and Rab7.*
- 4. In Discussion it is important to put this new mechanism in the context of what is already known about VEGFR2 endocytosis and its control- in particular effects of Ephrin B2 and Neuropilin-1*

Responses:

- 1. How specific is the ATP7A-VEGFR2 interaction? Does ATP7A ko (OR KD in vitro) affect degradation of other transmembrane proteins and in particular, receptor tyrosine kinases that are trafficked in a manner similar to VEGFR2 (e.g. IGF1R, Insulin receptor, etc.). and in manner distinct from VEGFR2 (e.g. FGFR1)?*

Response: We performed additional experiments and found that ATP7A knockdown (KD) had no significant effects on the degradation of Insulin receptor 1 β (IR1 β) or FGFR1 in response to insulin and FGF, respectively, in ECs (revised **Figure S3BC**). Thus, the effects of ATP7A which we report appear to be specific to VEGFR2. This issue has been included in the **Results** (page 8, the first paragraph) and is addressed in the **Discussion** (page 13, the second paragraph) in the revised manuscript.

- 2. Are there any effects of ATP7A KO on VEGFR3? Other VEGFR2 binding partners such as Nrp1?*

Response: We performed additional measurements and found that ATP7A KD had no significant effects on degradation of VEGFR3 in response to VEGF-C (**Figure S3D**) as well as the VEGFR2 binding partner Nrp1 expression in response to VEGF in HUVECs under conditions where it promoted ligand-induced VEGFR2 degradation (**Figure S3A**). This issue has

been included in the **Results** (page 8, the first paragraph) and the **Discussion** (page 13, the second paragraph and page 15, the first paragraph) in the revised manuscript.

3. *It is important to fully investigate all aspects of VEGFR2 trafficking. While Rab5 and Lamp2 data are convincing it would be good to provide data regarding the col-localization extent with Rab11 and Rab7.*

Response: We extended our studies and performed colocalization analysis for VEGFR2 with Rab11 or Rab7 in addition to Rab5 and Lamp2 in HUVECs to support the conclusion that ATP7A functions to limit lysosomal degradation of VEGFR2 in VEGF-stimulated ECs. This issue has now been included in the **Results** (page 9, second paragraph) and in revised **Figure 4C** in the revised manuscript.

4. *In Discussion it is important to put this new mechanism in the context of what is already known about VEGFR2 endocytosis and its control- in particular effects of Ephrin B2 and Neuropilin-1.*

Response: As the reviewer suggested, we have now included what is already known about VEGFR2 endocytosis and its control in particular effects of Ephrin B2 and Nrp-1” in the **Discussion** (page 14, the third paragraph, page 15, the first paragraph).

Response to Reviewer # 2

Comments:

This is a study that evaluates the role of ATP7A in angiogenesis via VEGFR2. The topic builds on the author's previous studies of ATP7A and angiogenesis and it is unclear how novel the current results are compared to previous work. I find shortcomings in the experimental parts aiming to reveal the role of ATP7A. For example, when using siRNA, the authors should check how much the protein expression is downregulated in the cells by this treatment. We have found that this can vary a lot and optimization in terms of both conc of RNA and time of incubation is needed. The authors also claim a direct interaction between VEGFR2 and ATP7A using co-IP. The data in Fig 3A is not convincing. Truly there are no controls and it could be nonspecific binding or a tiny fraction of no biological relevance. This needs to be further addressed. Co-IP experiments are very complicated and depends on many parameters. To claim protein-protein interaction, additional experiments are needed. This is the one key of the paper, as the authors claim to reveal mechanistic information in this study.

Also, the authors state that the effects/pathway is independent of Cu. This is suggested based on TTM experiments, a chelator that would coordinate Cu. However, it is not shown that TTM enters the cells nor that it chelates Cu in the cell such that it is not available to ATP7A. Assessing Cu-dependence is hard as Cu is involved in so many things in cells and modulating the Cu levels will necessarily affect also other cellular processes. This should be further investigated before a claim can be made. Also, adding excess Cu should be included, and analysis (ICPMS) of internal or cell-associated Cu at different conditions determined to show

that treatments have the expected effect. TTM is controversial as it can also bind to metal-binding domains in ATP7A via the Cu (forming a ternary complex).

Most of the experimental data differences have low significance, nothing better than one star and many changes noted are less than 2-fold. this is not highly convincing. also, for co-localization between ATP7A and VEGFR2, the performed experiments cannot determine true co-localization as in near each other such that interaction takes place. the resolution is too low. truly, this only shows that the two proteins are in the same area of cells. Also, likely these cells express ATP7B and parallel path involving ATP7B should be addressed (or excluded). ATP7B is supposed to mostly be in hepatic cells, but when one investigates most cells express some ATP7B.

Also, the antibody towards ATP7A needs to be verified as there are several and all do not work well. Taken together, it is hard to judge the significance of the message, but regardless, i find many shortcomings in the experimental approach that should be addressed.

Responses:

1. *This is a study that evaluates the role of ATP7A in angiogenesis via VEGFR2. The topic builds on the author's previous studies of ATP7A and angiogenesis and it is unclear how novel the current results are compared to previous work.*

Response: We respectfully suggest that our study provides clear evidence of unexpected crosstalk between the Cu transporter ATP7A, the process of autophagy and VEGFR2 degradation. The current study provides the first evidence that VEGF stimulation promotes ATP7A binding to VEGFR2, which then limits lysosomal/autophagy-mediated VEGFR2 degradation in ECs in a Cu-independent (TTM-insensitive) manner. The canonical role of ATP7A is to deliver Cu to the secretory Cu-dependent enzymes including SOD3 and LOX to regulate its activity^{1,2}. We and others showed that LOX promotes tumorigenesis and metastasis³ and is partially involved in VEGF-induced angiogenesis in ECs^{4,5}. We reported that SOD3 is required for post-ischemic neovascularization⁵. These results suggest that the Cu-LOX axis-dependent ATP7A-mediated angiogenesis may be downstream of ATP7A-mediated Cu-independent regulation of VEGFR2 signaling. Alternatively, ATP7A binding to VEGFR2, in addition to Cu-dependent ATP7A-mediated LOX activity, in turn, enhances VEGF-induced angiogenesis in ECs. In addition, in the present study, we have developed EC-specific ATP7A KO mice by crossing ATP7A-floxed mice with tamoxifen-inducible Cdh5-Cre-ERT2 mice and demonstrated that endothelial ATP7A plays an essential role in post-ischemic angiogenesis and neovascularization *in vivo*. These findings are novel and, to our knowledge have not previously been reported. We have emphasized this point in the **Abstract**, **Discussion** (page 13, the first paragraph, page 14 first and second paragraph) in the revised manuscript.

2. *I find shortcomings in the experimental parts aiming to reveal the role of ATP7A. For example, when using siRNA, the authors should check how much the protein expression is downregulated in the cells by this treatment. We have found that this can vary a lot and optimization in terms of both conc of RNA and time of incubation is needed.*

Response: This is an important point and we apologize for omitting several Western blots showing ATP7A protein knockdown in siATP7A-transfected ECs in our original figures. This

omission has been corrected in the revised figures. When we initiated this study, we optimized the concentrations and incubation times of ATP7A siRNA in HUVECs. We also included other ECs, such as bovine aortic ECs (BAEC), showing the efficient knockdown of ATP7A proteins using two different ATP7A siRNAs, which demonstrated enhanced ligand-induced VEGFR2 downregulation (revised **Figure S2**). We respectfully suggest that role of ATP7A in limiting autophagic VEGFR2 degradation to promote angiogenesis in ECs is now convincingly demonstrated and, to our knowledge, is novel.

3. *The authors also claim a direct interaction between VEGFR2 and ATP7A using co-IP. The data in Fig 3A is not convincing. Truly there are no controls and it could be nonspecific binding or a tiny fraction of no biological relevance. This needs to be further addressed. Co-IP experiments are very complicated and depends on many parameters. To claim protein-protein interaction, additional experiments are needed. This is the one key of the paper, as the authors claim to reveal mechanistic information in this study.*

Response: This is also an important point, and to address these doubts we have significantly extended our experimental evidence. We have now addressed this issue by including the negative control for co-IP (IP with IgG) in the revised Figure 3A. Furthermore, to monitor *in situ* formation of ATP7A-VEGFR2 complex, we have used the Duolink proximity ligation assay (PLA)⁶, where red puncta indicate positive staining that the epitopes of the target proteins are in close proximity (<40 nm). We found that both primary mouse ATP7A antibody and rabbit VEGFR2 antibody combined with secondary PLA probes dramatically increased the PLA positive red dots after VEGF stimulation in HUVECs (Figure 3B). By contrast, single primary antibody only or IgG negative control did not display red dots (Figure 3B). To confirm further the direct interaction of VEGFR2 and ATP7A, we also performed “co-transfection analysis” of myc-ATP7A and flag-VEGFR2 plasmids in Cos7 cells and found that myc-ATP7A, but not IgG, was co-immunoprecipitated with flag-VEGFR2 after VEGF stimulation (revised **Figure 3C**). Taken together, these results strongly support our finding that ATP7A binds to VEGFR2 after VEGF stimulation. These issues have been included in the **Results** (page 8, the last paragraph and page 9 the first paragraph), the **Discussion** (page 13, the second paragraph) in the revised manuscript.

4. *Also, the authors state that the effects/pathway is independent of Cu. This is suggested based on TTM experiments, a chelator that would coordinate Cu. However, it is not shown that TTM enters the cells nor that it chelates Cu in the cell such that it is not available to ATP7A. Assessing Cu-dependence is hard as Cu is involved in so many things in cells and modulating the Cu levels will necessarily affect also other cellular processes. This should be further investigated before a claim can be made. Also, adding excess Cu should be included, and analysis (ICPMS) of internal or cell-associated Cu at different conditions determined to show that treatments have the expected effect. TTM is controversial as it can also bind to metal-binding domains in ATP7A via the Cu (forming a ternary complex).*

Response: We agree with Reviewer #2 regarding the potentially complex and multiple actions of tetrathiomolybdate, TTM. TTM has undergone several clinical trials and is now used as a “decoppering agent” in the treatment of Wilson Disease and has been tested in a number of trials as an anti-tumor agent^{7, 8, 9, 10}. It is clear that its biological effects are complicated and

some of the effects in Wilson Disease patients may well be due to both intra and extracellular effects of TTM. TTM is known to be an efficient Cu chelator and its affinity for Cu(I) has been reported, and is $2.32 \times 10^{-20} \text{M}^{-1}$, these authors also reported on its ability to demetallate Cu, Zn SOD. It is also known to inhibit copper trafficking proteins through metal cluster formation. That is, forming, a ternary complex with Cu chaperone Atox1, the structure of which was recently determined¹². It also forms similar ternary complexes with ATP7A and other Cu-binding proteins¹³. It is well-established that TTM is “a membrane-permeable Cu chelator”¹⁴ as its presence altered intracellular ATP7A trafficking (without added extracellular Cu), and it has also been shown to inhibit oxygen consumption by mitochondria in several cell lines (see for example¹⁵, among others). These observations are difficult to interpret if TTM does not have access to the intracellular compartment. All of these observations make it difficult to provide simple mechanistic explanations for cellular effects of TTM. However, we would like to emphasize that the addition of TTM, in our system has no effect on the siATP7A-induced degradation of VEGFR2 (see our Figure 2H).

Moreover, to demonstrate the efficacy of TTM in our system, we showed that TTM almost completely inhibited VEGF-induced Cu dependent lysyl oxidase (LOX) activity (revised **Figure S4B**) as well as CuCl_2 -induced phosphorylation of ERK, which is downstream of MEK which is directly activated by Cu^{16,17} (revised **Figure S4C**). By contrast, TTM had no effect on either ATP7A depletion-induced VEGFR2 degradation or inhibition of VEGFR2 downstream signaling. These findings suggest that TTM, as used in this study chelates intracellular Cu effectively and that ATP7A KD-mediated ligand-induced VEGFR2 degradation is Cu - independent. These issues have been included in the **Results** (page 8, the second paragraph) and are now discussed in the **Discussion section** (page 18, the first paragraph) in the revised manuscript.

5. *Most of the experimental data differences have low significance, nothing better than one star and many changes noted are less than 2-fold. this is not highly convincing.*

Response:

We respectfully suggest that small differences if statistically significant do not have to be large to have biological significance. Indeed, in the original figures, we expressed significance $* < 0.05$ by including < 0.01 and < 0.001 . We have now revised several figures by changing $* < 0.05$ to $** < 0.01$ and $\# < 0.001$ by re-analyzing the data and by repeating experiments in the revised manuscript.

6. *also, for co-localization between ATP7A and VEGFR2, the performed experiments cannot determine true co-localization as in near each other such that interaction takes place. The resolution is too low. truly, this only shows that the two proteins are in the same are of cells.*

Response: We agree with the reviewer that a co-localization study does not, by itself, provide direct evidence that ATP7A interacts with VEGFR2 in ECs. Thus, as described above in the response No.3, we performed additional experiments using “Proximity Ligation Assay (PLA) assay” (revised **Figure 3B**) and “Co-transfection Assay” (revised **Figure 3C**) to show the direct association of VEGFR2 and ATP7A in a ligand-dependent manner. Furthermore, we have shown

the co-localization of ATP7A and VEGFR2 after VEGF stimulation using confocal microscopy, in the yellow fluorescence in the merged images (analyzed by fluorescence intensity/distance for each protein) (see revised **Figure 3D**), as we have reported¹⁸. This issue has been included in the **Results** (page 8, the last paragraph and page 9 the first paragraph), and in the **Discussion** (page 13, the second paragraph) in the revised manuscript.

7. *Also, likely these cells express ATP7B and parallel path involving ATP7B should be addressed (or excluded). ATP7B is supposed to mostly be in hepatic cells, but when one investigates most cells express some ATP7B.*

Response: To address the reviewer's concern, we examined the protein expression of ATP7B in various ECs including HUVEC, MAEC (mouse aortic ECs), BAEC (bovine aortic ECs) as well as mouse aorta and liver (positive control) using Western blot analysis. As shown in **Figure I** below, ATP7B protein was not expressed or not detectable in ECs or aorta except liver. Thus, the present study focused on investigating a role of ATP7A in VEGF-induced angiogenesis in ECs.

Figure I: ATP7B is expressed in liver but not in ECs or aorta. Lysates from HUVECs, MAECs, BAECs, mouse aorta and mouse liver were used for western blots using anti-ATP7B antibody.

8. *Also, the antibody towards ATP7A needs to be verified as there are several and all do not work well.*

Response: To verify the specificity of ATP7A antibody used in this study (LifeSpan BioSciences, catalog number#LS-B8162, Lot#44771), we measured the ATP7A protein level in HUVECs transfected with specific ATP7A siRNA. We re-ran the remaining lysates to show that expression of ATP7A protein, but not actin, was markedly decreased by ATP7A siRNA compared to control siRNA, and show these data in the revised manuscript. In addition, using BAECs transfected with 2 different siRNAs which target different regions of ATP7A mRNA, we found that both ATP7A siRNAs markedly decreased ATP7A protein expression and reduced VEGFR2 protein expression in ECs after VEGF stimulation (revised **Figure S2**). Thus, we respectfully suggest that we have validated the ATP7A antibody used in our study. We do recognize that antibody quality and efficiency may also vary with the lot.

Response to Reviewer # 3

Comments:

This paper is obviously innovative. The main role of ATG7A originally is to maintain the copper homeostasis in cells. Copper ions are known to promote angiogenesis. This research found that ATP7A binds to VEGFR2 in ECs in response to VEGF and functions to protect against p62/SQSTM1 mediated autophagic and lysosomal degradation of VEGFR2 and promote

VEGFR2 signaling in a Cu-independent manner, thereby promoting angiogenesis. The logic of this paper is clear.

However, the paper has several disadvantages:

(1) The paper proves that ATP7A can affect angiogenesis. But the authors do not prove that VEGF-induced EC angiogenesis in HUVECs transfected with ATP7A siRNA. Demonstrating angiogenesis in vitro could make the study more adequate.

(2) It is known that the UBA domain of p62 interacts with ubiquitin and ubiquitin-labeled proteins and acts as a selective autophagy receptor, degrading the labeled proteins. This paper proved that ATP7A inhibits the ubiquitination of VEGFR, and p62 can promote the ubiquitination of VEGFR through UBA, and ATP7A can inhibit the interaction of p62 and VEGFR, but there is no proof on how ATP7A works through p62-UBA. If the problem is proved, the research content will be more in-depth.

(3) The pictures of Western blot are not clear, such as Fig2.D, Fig5.C;

(4) The figures are not arranged neatly

Responses:

1. *The paper proves that ATP7A can affect angiogenesis. But the authors do not prove that VEGF-induced EC angiogenesis in HUVECs transfected with ATP7A siRNA. Demonstrating angiogenesis in vitro could make the study more adequate.*

Response: The process of angiogenesis includes EC migration and capillary tube formation. In our original manuscript, we showed that ATP7A knockdown using siRNA in HUVECs significantly inhibited VEGF-induced EC migration using the modified Boyden chamber assay and wound scratch assay, as well as capillary tube formation using spheroid EC sprouting assay in **original Figure 2B and 2C**. In addition, we also showed that VEGF-induced EC migration is also inhibited in ECs isolated from ATP7A dysfunctional ATP7A mutant mice compared to control mouse ECs (**original Figure 2A**). Thus, we believe that our study provides compelling evidence that ATP7A is required for VEGF-induced angiogenesis in ECs. This issue has been included in the **Discussion** (page 14, the second paragraph) in the revised manuscript.

2. *It is known that the UBA domain of p62 interacts with ubiquitin and ubiquitin-labeled proteins and acts as a selective autophagy receptor, degrading the labeled proteins. This paper proved that ATP7A inhibits the ubiquitination of VEGFR, and p62 can promote the ubiquitination of VEGFR through UBA, and ATP7A can inhibit the interaction of p62 and VEGFR, but there is no proof on how ATP7A works through p62-UBA. If the problem is proved, the research content will be more in-depth.*

Response: We appreciate these points raised by the reviewer and have attempted to address them in our revised manuscript. To address relationship between ATP7A and p62, containing the UBA domain, we examined their interaction but found that they did not interact with each other (**Figure S6**). Instead, we found that ATP7A directly interacts with VEGFR2 after VEGF stimulation using co-IP (**Figure 3A**), proximity ligation assay (PLA)(revised **Figure 3B**), and co-transfection of HA-VEGFR2 and myc-ATP7A (revised **Figure 3C**). Furthermore, we performed additional experiments to demonstrate that ATP7A knockdown-induced p62 binding to VEGFR2 as well as VEGFR2 down-regulation in VEGF-stimulated ECs were rescued by re-

expression of myc-hATP7A (revised **Figure 6D**). Taken together, these results suggest that ATP7A binding to VEGFR2 may protect VEGFR2 against degradation by preventing p62 binding to VEGFR2. This issue has been included in the **Results** (page 8, the last paragraph and page 9 the first paragraph) and the **Discussion** (page 17, the first paragraph) in the revised manuscripts.

3. *The pictures of Western blot are not clear, such as Fig2.D, Fig5.C*

Response: We apologize for this. We have replaced the western blots in original Figure 2D and 5C and now present them in revised Figure 2D and Figure 6C in the revised manuscript.

4. *The figures are not arranged neatly.*

Response: Thank you for the comments. We have rearranged the figures with additional data.

References

1. Fukai T, Ushio-Fukai M, Kaplan JH. Copper transporters and copper chaperones: roles in cardiovascular physiology and disease. *Am J Physiol Cell Physiol* **315**, C186-C201 (2018).
2. Lutsenko S, Barnes NL, Bartee MY, Dmitriev OY. Function and regulation of human copper-transporting ATPases. *Physiological reviews* **87**, 1011-1046 (2007).
3. Shanbhag V, *et al.* ATP7A delivers copper to the lysyl oxidase family of enzymes and promotes tumorigenesis and metastasis. *Proc Natl Acad Sci U S A* **116**, 6836-6841 (2019).
4. Chen GF, *et al.* Copper Transport Protein Antioxidant-1 Promotes Inflammatory Neovascularization via Chaperone and Transcription Factor Function. *Sci Rep* **5**, 14780 (2015).
5. Kim HW, Lin A, Guldberg RE, Ushio-Fukai M, Fukai T. Essential role of extracellular SOD in reparative neovascularization induced by hindlimb ischemia. *Circulation research* **101**, 409-419 (2007).
6. Söderberg O, *et al.* Direct observation of individual endogenous protein complexes in situ by proximity ligation. *Nat Methods* **3**, 995-1000 (2006).
7. Pan Q, *et al.* Copper deficiency induced by tetrathiomolybdate suppresses tumor growth and angiogenesis. *Cancer research* **62**, 4854-4859 (2002).
8. Brewer GJ. Copper control as an antiangiogenic anticancer therapy: lessons from treating Wilson's disease. *Experimental biology and medicine (Maywood, NJ)* **226**, 665-673 (2001).
9. Brewer GJ. Tetrathiomolybdate anticopper therapy for Wilson's disease inhibits angiogenesis, fibrosis and inflammation. *Journal of cellular and molecular medicine* **7**, 11-20 (2003).
10. Brewer GJ. Anticopper therapy against cancer and diseases of inflammation and fibrosis. *Drug discovery today* **10**, 1103-1109 (2005).

11. Smirnova J, *et al.* Copper(I)-binding properties of de-coppering drugs for the treatment of Wilson disease. α -Lipoic acid as a potential anti-copper agent. *Scientific reports* **8**, 1463 (2018).
12. Alvarez HM, *et al.* Tetrathiomolybdate inhibits copper trafficking proteins through metal cluster formation. *Science (New York, NY)* **327**, 331-334 (2010)
13. Fang T, *et al.* Tetrathiomolybdate induces dimerization of the metal-binding domain of ATPase and inhibits platination of the protein. *Nature communications* **10**, 186 (2019).
14. White C, *et al.* Copper transport into the secretory pathway is regulated by oxygen in macrophages. *J Cell Sci* **122**, 1315-1321 (2009).
15. Navrátilová J, *et al.* Selective elimination of neuroblastoma cells by synergistic effect of Akt kinase inhibitor and tetrathiomolybdate. *Journal of cellular and molecular medicine* **21**, 1859-1869 (2017).
16. Turski ML, *et al.* A novel role for copper in Ras/mitogen-activated protein kinase signaling. *Molecular and cellular biology* **32**, 1284-1295 (2012).
17. Brady DC, *et al.* Copper is required for oncogenic BRAF signalling and tumorigenesis. *Nature* **509**, 492-496 (2014).
18. Kim YM, *et al.* Redox Regulation of Mitochondrial Fission Protein Drp1 by Protein Disulfide Isomerase Limits Endothelial Senescence. *Cell Rep* **23**, 3565-3578 (2018).

REVIEWERS' COMMENTS

Reviewer #1 (Remarks to the Author):

Issues raised in the initial review have been adequately addressed.

Reviewer #2 (Remarks to the Author):

Many new experiments have been done and new data added in. The revisions are satisfactory. Thank you.

Reviewer #3 (Remarks to the Author):

I thank the authors for having clearly addressed all the issues presented in my previous revision. I believe a much better and improved version of the original work is now presented and could be considered for publication.